# DynQual v1.0: A high-resolution global surface water quality model

Edward R. Jones[1,*], Marc F.P Bierkens[1,2], Niko Wanders[1], Edwin H. Sutanudjaja[1], Ludovicus P.H van Beek[1], Michelle T.H. van Vliet[1]

[1]Department of Physical Geography, Faculty of Geosciences, Utrecht University, Utrecht, The Netherlands.

[2]Deltares, Unit Soil and Groundwater Systems, Utrecht, The Netherlands

* Correspondence to: e.r.jones@uu.nl

## Abstract

Maintaining good surface water quality is crucial to protect ecosystem health and for safeguarding human water use activities. Yet, our quantitative understanding of surface water quality is mostly predicated upon observations at monitoring stations that are highly limited in space and fragmented across time. Physically-based models, based upon pollutant emissions and subsequent routing through the hydrological network, provide opportunities to overcome these shortcomings. To this end, we have developed the dynamical surface water quality model (DynQual) for simulating water temperature (Tw) and concentrations of total dissolved solids (TDS), biological oxygen demand (BOD) and fecal coliform (FC) with a daily timestep and at 5 arc-minute (~10km) spatial resolution. Here, we describe the main components of this new global surface water quality model and evaluate model performance against in-situ water quality observations. Furthermore, we describe both the spatial patterns and temporal trends in TDS, BOD and FC concentrations for the period 1980–2019, also attributing the dominant contributing sectors to surface water pollution. Modelled output indicates that multi-pollutant hotspots are especially prevalent across northern India and eastern China, but that surface water quality issues exist across all world regions. Trends towards water quality deterioration have been most profound in the developing world, particularly Sub-Saharan Africa and southern Asia. The model code is available open-source (https://github.com/UU-Hydro/DYNQUAL) and we provide global datasets of simulated hydrology, Tw, TDS, BOD and FC at 5 arc-minute resolution with a monthly timestep (https://doi.org/10.5281/zenodo.7139222). This data has potential to inform assessments in a broad range of fields, including ecological, human health and water scarcity studies.

## 1. Introduction

Maintaining good surface water quality is important for protecting ecosystem health and ensuring human access to safe water resources for a diverse range of sectoral needs (Van Vliet et al., 2021; Jones et al., 2022). For example, high organic pollution can reduce oxygen availability and can lead to the suffocation of aquatic organisms (Sirota et al., 2013), while pathogen pollution represents a potential health risk for people exposed to this water. The consumption of contaminated drinking water can lead to the transmission of diseases such as cholera, dysentery and polio leads, which cause an estimated 485,000 deaths annually (Prüss-Ustün et al., 2019). Another example is salinisation of water resources, which can both limit irrigation water use (Thorslund et al., 2022) and threaten freshwater biodiversity (Velasco et al., 2019) where species cannot tolerate elevated salinity concentrations. Similarly, increased water temperatures can disrupt energy production (Van Vliet et al., 2016), while also providing more favourable conditions for cyanobacterial blooms that can lead to hypoxia which can disrupt freshwater habitats (Smucker et al., 2021).

Human activities, both directly and indirectly, cause changes in surface water quality relative to ambient ('pristine') conditions. Indirectly, altered precipitation patterns and the increased frequency of hydro-meteorological extremes that result from human-induced climate change can lead to fundamental changes in the hydrological regime (Wanders and Wada, 2015; Gudmundsson et al., 2021). Lower water levels due to altered seasonality patterns or droughts reduce the stream dilution capacity, which can increase the proportion of streamflow originating from (polluted) point sources (Wright et al., 2014; Luthy et al., 2015; Ehalt Macedo et al., 2022). Both of these factors increase river water contamination, threatening both the safe usability of water and environmental health. Climate change is also altering the thermal regime of rivers (Van Vliet et al., 2013), with higher temperatures also causing dissolved oxygen depletion (Ozaki et al., 2003).

More directly, sectoral activities generate return flows - water that is extracted for a specific purpose but is not consumed (evaporated) in the process – but which has changed in composition as a result of the water use activity (Sutanudjaja et al., 2018; Jones et al., 2021). For example, the composition of domestic wastewaters will reflect the various household water uses, including organic and fecal contamination from human waste (Wwap, 2017) and elevated nutrient concentrations from household chemicals and laundry detergents (Van Puijenbroek et al., 2019). The re-introduction of these flows back to the environment represent a significant source of pollutant loadings that degrade river water quality (Jones et al., 2022). Collection and treatment of these flows, before their re-introduction to the environment, can help to minimise the impact on surface water quality (Jones et al., 2022). Yet, these processes can be economically expensive to establish and operate, and hence collection and treatment infrastructure are not ubiquitous worldwide (Jones et al., 2021; Jones et al., 2022).

Water quality is an integral part of the Sustainable Development Agenda, cross-cutting almost all Sustainable Development Goals (SDGs). Despite widespread recognition of its importance, water quality monitoring data is still severely lacking in several world regions – particularly Africa and Central Asia (Damania et al., 2019). Furthermore, in regions where observation data is available, data is often sparse in both space and time. Water quality models offer opportunities to overcome these limitations (Hofstra et al., 2013; Beusen et al., 2015; UNEP, 2016; Van Vliet et al., 2021). As opposed to statistical models which heavily rely on observed water quality data, physical models simulate the emission and transport of pollutant loadings along the river network directly based on climatic, hydrological and socio-economic input data. This makes physically-based model approaches especially advantageous when simulating water quality in ungauged catchments and for projecting water quality under future (uncertain) climatic and socio-economic developments (Wanders et al., 2019).

A spatially and temporally detailed assessment of multiple water quality constituents at the global scale is lacking. Furthermore, only a few studies have quantitatively evaluated temporal dynamics and

trends in water quality over extended time periods, particularly considering changes in factors that drive higher pollutant emissions (e.g. population growth, industrialisation) relative to factors that abate pollutant emissions (e.g. wastewater treatment). Lastly, few studies have assessed the spatio-temporal patterns in the specific sectoral activities that are driving patterns in surface water quality worldwide.

Here, we present a high-spatiotemporal resolution surface water quality model (henceforth *DynQual*), which can currently be used to simulate water temperature (Tw), and concentrations of total dissolved solids (TDS) to represent salinity pollution, biological oxygen demand (BOD) to represent organic pollution and fecal coliform (FC) as a coarse indicator for pathogen pollution. All simulations are provided at a daily timestep with a spatial resolution of 5x5 arc-minutes (approx. 10km at the

equator). DynQual considers a wide range of hydro-climatic and socio-economic drivers, spanning across the major contributing pollutant sources. The high spatio-temporal resolution of DynQual, combined with these features, allows the model to address scientific questions that are not currently possible using existing surface water quality models. For example, while previous work has compared pollutant loads (masses) originating from different sources at aggregated spatial scales (i.e. basin or

subbasin level), the impact on in-stream concentrations - which is also dependent upon spatio-temporal variability in dilution capacity and in-stream decay processes – has not been assessed.

    The objectives of this study are to: 1) introduce a new open-source global surface water quality model and evaluate model performance; 2) assess spatial patterns and trends in surface water quality, focussing on total dissolved solids (TDS), biological oxygen demand (BOD) and fecal coliform (FC)

concentrations for the period 1980 – 2019; and 3) demonstrate additional model capabilities by assessing the sector-specific contributions towards surface water pollution across both space and time.

## 2. Model description

### 2.1 General overview

The newly developed DynQual model builds on the modelling framework of DynWat, a global water temperature model that solves the energy-water balance to simulate daily water temperature (Tw) and ice thickness (Van Beek et al., 2012; Wanders et al., 2019). A full model description including the energy balance equations and the representation of ice cover, floodplains, channel roughness and lakes and reservoirs within DynWat is available in published literature (Wanders et al., 2019).

DynQual further includes the impact of heat dumps produced in thermo-electric powerplants (Van Vliet et al., 2012a; Van Vliet et al., 2021) on water temperature. In addition to water temperature, DynQual simulates daily in-stream concentrations of three water quality constituents, namely, total dissolved solids (TDS), biological organic matter (BOD) and fecal coliform (FC), which are of key social and environmental relevance (Van Vliet et al., 2021) (Figure 1).

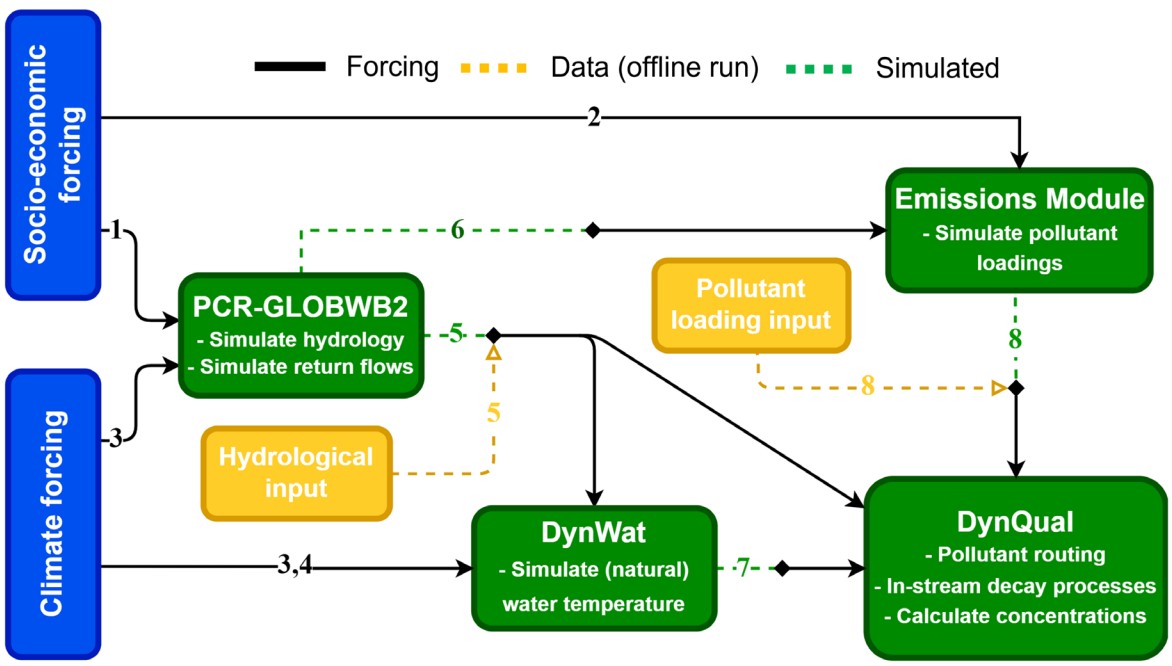

**Socio-economic forcing**
1. Domestic gross and net water demand (m day⁻¹); industry gross and net water demand (m day⁻¹); livestock gross and net water demand (m day⁻¹)
2. Population numbers; livestock numbers; urban fraction; thermo-electric power return flows (m³ s⁻¹); excretion rates (g or cfu day⁻¹); mean effluent concentrations (mg l⁻¹ or cfu 100ml⁻¹)

**Climate forcing**
3. Air temperature (°C); precipitation (m day⁻¹); potential evapotranspiration (m day⁻¹)
4. Solar radiation (W m⁻²); annual average air temperature (°C); vapour pressure (kPa); cloud cover (%); sunlight hours (hours day⁻¹)

**Hydrological data**
5. Baseflow or groundwater discharge (m day⁻¹); interflow (m day⁻¹); direct or surface runoff (m day⁻¹)
6. Manufacturing return flows (m³ day⁻¹); irrigation return flows (m³ day⁻¹); urban surface runoff (m³ day⁻¹)
7. Water temperature (°C)

**Pollutant loading data**
8. Anthropogenic temperature loading (MW); TDS loading (g day⁻¹); BOD loading (g day⁻¹); FC loading (10⁶ cfu day⁻¹)


**Figure 1.** Overview of the required input data for running DynQual in different model configurations. Runs coupled with PCR-GLOBWB2 require socio-economic (arrow 1) and climatic forcing (3,4) data as standard, with options to either 1) estimate loads based on additional socio-economic (2) and simulated hydrological (6) data; or 2) provide pollutant loadings directly as input data (8). Offline
runs require both hydrological (5) and pollutant loadings (8) input data to be provided directly.

We also offer two options for running DynQual, either: 1) in a stand-alone configuration with specific discharge (i.e. baseflow, interflow and direct runoff in m day$^{-1}$) fed from any land surface or hydrological model, or 2) coupled with the global hydrological and water resources model PCR-GLOBWB2 (Sutanudjaja et al., 2018). The routine for surface water (and pollutant) routing follows an eight-point steepest-gradient algorithm across the terrain surface (local drainage direction) in a convergent drainage network with the lowermost cell connected to either the ocean or an endorheic basin as per PCR-GLOBWB2 (Sutanudjaja et al., 2018) and DynWat (Van Beek et al., 2012; Wanders et al., 2019). Routing within DynQual uses the kinematic wave approximation of the Saint-Venant equations with flow described by Manning's equation, solved using a time-explicit variable sub-time stepping scheme based on the minimum Courant number (Sutanudjaja et al., 2018). In the coupled configuration, surface waters are subject to water withdrawals and return flows from the domestic, industrial, livestock and irrigation sectors calculated within the water use module of PCR-GLOBWB2. A complete model description of PCR-GLOBWB2 including detailed information on the model structure, individual modules (meteorology, land surface, groundwater, surface water routing and water use) and validation of hydrological output is available in published literature (Sutanudjaja et al., 2018). In both configurations of DynQual, pollutant loadings can be prescribed directly (akin to a forcing). Alternatively, when running DynQual coupled with PCR-GLOBWB2 pollutant loadings can be simulated within the model runs by providing only simple input data (SI Section 1). An overview of DynQual, which details the input data required for the different model configurations, is displayed (Figure 1). By providing these options, we allow for flexibility – allowing pollutant loadings to be directly imposed on the model facilitates users to estimate loadings using their preferred methodology and assumptions; whereas the option to estimate pollutant loadings within the model run enables users to simulate water quality without any pre-processing requirements but still provides flexibility to use their preferred input datasets. Parameter values related to pollutant emissions can be adjusted by the user, as desired. When simulating pollutant loadings within model runs, it is also possible to quantify the contribution and relative importance of different water use sectors to the spatial patterns and temporal trends in surface water quality.

As per PCR-GLOBWB2 (Sutanudjaja et al., 2018) and DynWat (Wanders et al., 2019), DynQual is written in Python 3 and is run using an initialization (.ini) file in which key aspects of the model run are defined (e.g. spatial extent, simulation period, paths to parameter and forcing files). Most input files required and all output files are in NetCDF format. Global 5 arc-min DynQual runs that are coupled with PCR-GLOBWB2 have a wall-clock time of approximately 6 hours per year when run with parallelisation, due to the requirement to use the kinematic wave routing option for higher accuracy discharge and water temperature simulations. This is approximately equivalent to the PCR-GLOBWB2 run times given by Sutanudjaja et al., (2018). DynQual runs performed in the stand-alone configuration are faster (~20%).

## 2.2 Water quality equations

### 2.2.1 Water temperature (Tw)

Water temperature (Tw) is simulated by solving the surface water energy balance, using the DynWat model as basis (Van Beek et al., 2012; Wanders et al., 2019). In addition to solving the surface water energy balance, DynWat also accounts for surface water abstraction, reservoirs, riverine flooding and the formation of ice (Wanders et al., 2019). Here, we further develop DynWat to include advected heat flows from thermo-electric powerplants, as per the method described in van Vliet et al., (2012; 2016). The modelling equations for Tw incorporated into DynQual are shown in Eq. [1] and are fully elaborated on in previous work (Van Beek et al., 2012; Van Vliet et al., 2012a; Van Vliet et al., 2016; Wanders et al., 2019).

$$\rho_w C_p \frac{\partial (hT_w)}{\partial t} = \rho_w C_p \frac{\partial (vT_w)}{\partial x} + H_{tot} + \rho_w C_p \int_{x=0}^{dx} q_s T_s + \frac{Tw_{pow_n}}{h * w * dx}$$

$$H_{tot} = S_{in}(1 - a_w) + L_{in} - L_{out} - H - LE$$

$$Tw_{pow_n} = \rho_w * C_p * RF_{pow,n} * \Delta T_{pow\_rf}$$

[1]

Where $t$ is time , $x$ is location along the drainage network, $T_w$ is water temperature (K), $C_p$ is the specific heat capacity of water (4,190 J kg$^{-1}$ K$^{-1}$), $\rho_w$ is the density of fresh water (1000 kg m$^{-3}$), $h$ is the stream water depth (m), $v$ is the velocity of water (m s$^{-1}$), $H_{tot}$ is the heat flux at the air-water interface, $S_{in}$ is the incoming shortwave radiation (J m$^{-2}$ s$^{-1}$), $1 - a_w$ is the reflected shortwave radiation (J m$^{-2}$ s$^{-1}$), $L_{in}$ is the incoming longwave radiation (J m$^{-2}$ s$^{-1}$), $L_{out}$ is the outgoing longwave radiation (J m$^{-2}$ s$^{-1}$), $H$ is the sensible heat flux (J m$^{-2}$ s$^{-1}$), LE is the latent heat flux (J m$^{-2}$ s$^{-1}$), $q_s$ are the lateral water fluxes from land to stream (m s$^{-1}$), $T_s$ is the temperature of lateral water fluxes (K), $Tw_{pow_n}$ is the heat dump from thermo-electric powerplants (J s$^{-1}$), $RF_{pow}$ is the return flows of cooling water from thermo-electric powerplants (m$^3$ s$^{-1}$), $\Delta T_{pow\_rf}$ is the difference in water temperature between the return flows and ambient river water (K), $w$ is the stream width (m) and $dx$ is the distance between gridcell $n$ and the upstream gridcell $n$-$1$ (m).

### 2.2.2   Conservative (TDS) and non-conservative (BOD, FC) substances

Our modelling strategy for total dissolved solids (TDS), biological oxygen demand (BOD) and fecal coliform (FC) is a mass balance approach assuming transport by advection only, whereby sector-specific loadings (i.e. masses of pollutants generated from a particular human activity in a given time period) are accumulated from all contributing sectors and routed through the global stream network until outflow to the ocean or an endorheic basin (Thomann and Mueller, 1987; Chapra and Pelletier, 2004; Voß et al., 2012; UNEP, 2016; Van Vliet et al., 2021).

TDS is modelled as a conservative substance, while BOD and FC are modelled as non-conservative substances that include first-order decay processes (Voß et al., 2012; Reder et al., 2015; UNEP, 2016; Van Vliet et al., 2021). Our approach for both the conservative and non-conservative substances assumes instantaneous and full mixing of all streamflow and return flows in each grid cell. As per most water quality models, DynQual simulates water quality per individual gridcell over a consecutive series of discrete time periods (Loucks and Beek, 2017). Each gridcell represents a volume element, which is in steady-state conditions within each time period, which also contains a (fully-mixed) pollutant mass (Figure 2). In each consecutive timestep, there is an associated volume of water and mass of pollutant that flows into the gridcell from upstream and that flows out of the gridcell to the downstream gridcell. For non-conservative substances, there are also gridcell-specific in-stream decay processes that influence the total mass of pollutant in each sub-time interval. DynQual simulates these transport and decay processes with a sub-daily interval ($\Delta t$ in seconds), the length of which is determined with respect to channel characteristics and discharge (SI Section 2 & SI Eq. [9]).

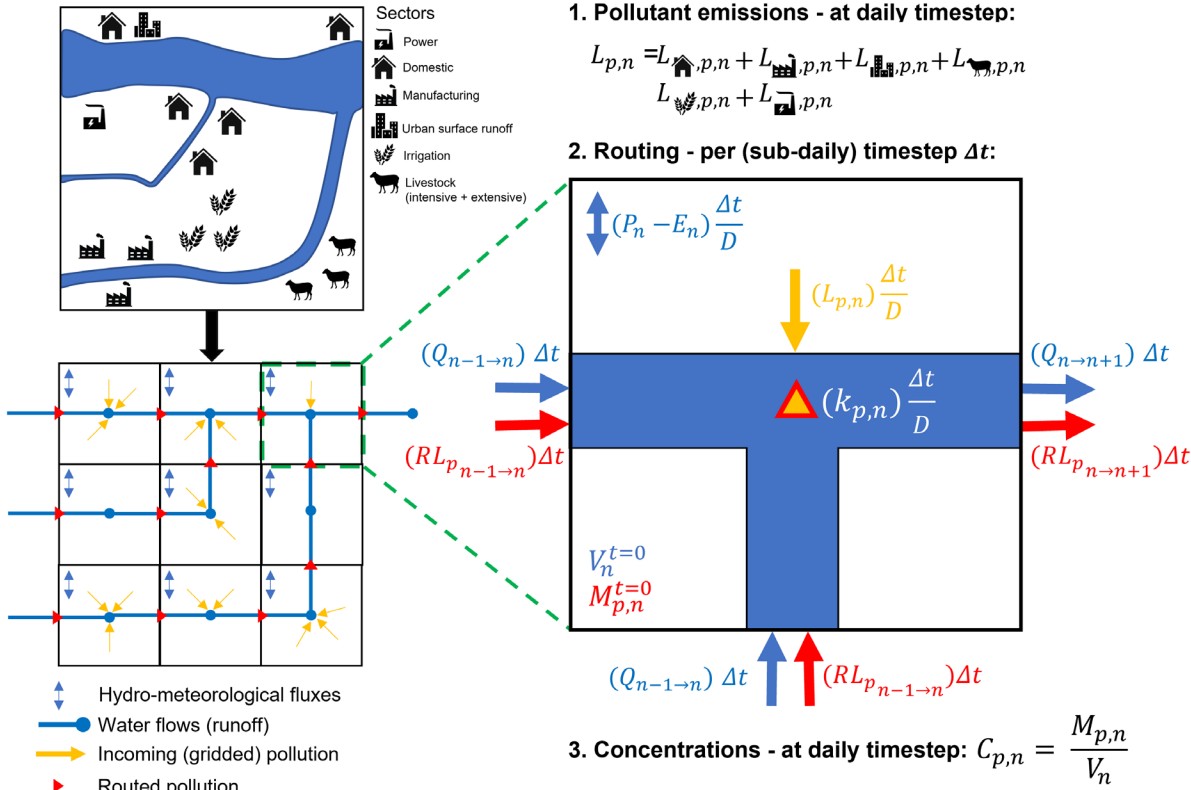

**Figure 2.** Schematic overview of DynQual, including a translation of local hydrological and socio-economic situation (a) into a local drain direction (LDD) map that includes hydrological and pollutant fluxes (b) and a representation of the gridcell based processes (pollutant emission calculation, routing procedure and computation of pollutant concentrations) in an individual DynQual gridcell (c). $C_{p,n}$ is the concentration of pollutant $p$ (e.g. mg l$^{-1}$), while $M_{p,n}$ is the total mass of pollutant $p$ (e.g. g) and $V_n$ is the channel storage (m$^3$), all in gridcell $n$. $V_n^{t=0}$ is the volume of channel storage from the previous timestep (m$^3$), while $Q_{n-1\to n}$ and $Q_{n\to n+1}$ is the discharge (m$^3$ s$^{-1}$) into and out of gridcell $n$, respectively, per timestep $\Delta t$. $M_{p,n}^{t=0}$ is the mass of pollutant $p$ from the previous timestep, while $RL_{p_{n-1\to n}}$ and $RL_{p_{n\to n+1}}$ are the loadings of pollutant $p$ (e.g. g s$^{-1}$) that are routed into and out of gridcell $n$, respectively, per timestep $\Delta t$. $L_{p,n}$ are the combined local loadings of pollutant $p$ (e.g. g day$^{-1}$) in gridcell $n$, which is the sum of loadings from all contributing sectors and urban surface runoff. $k_{p,n}$ represents a decay coefficient, which depends upon pollutant $p$ (-). $D$ is the length of a day in seconds (i.e. 86 400 s day$^{-1}$), while $\Delta t$ is the length of the sub-timestep (s) which is linked to the internal routing regime within DynQual & PCR-GLOBWB2. $P_n$ is precipitation (m$^3$ day$^{-1}$) and $E_n$ is evapotranspiration (m$^3$ day$^{-1}$), with these terms included as an example of gridcell-specific hydrological fluxes. For a more detailed overview of the hydrological fluxes within a gridcell we refer to the PCR-GLOBWB 2 documentation (Sutanudjaja et al., 2018).

The pollutant concentration at each subsequent time interval $(t + \Delta t)$ is calculated following Eq. [2]. It should be noted that, while we simulate the terms of this equation with a sub-daily timestep interval, DynQual only reports concentrations in the final sub-daily interval of each day. This is due to the lack of sub-diurnal input data, for efficient data storage and the lack of relevance of such high-resolution simulations with respect to our large-scale modelling approach.

$$C_{p,n}^{t+\Delta t} = \frac{M_{p,n}^{t+\Delta t}}{V_n^{t+\Delta t}} + BG_{i,n}$$

[2]

Where $C_{p,n}^{t+\Delta t}$ and $M_{p,n}^{t+\Delta t}$ is the concentration and mass, respectively, of pollutant $p$ in gridcell $n$ at the consecutive time interval $(t+\Delta t)$, whereas $V_n^{t+\Delta t}$ is the volumetric channel storage (m$^3$) in this gridcell in the same interval. $V_n^{t+\Delta t}$ is simulated directly within PCR-GLOBWB2, accounting for the initial storage, discharge into and out of gridcell $n$ over the time interval $\Delta t$ and gridcell specific hydrological fluxes including precipitation and evapotranspiration (Sutanudjaja et al., 2018). $M_{p,n}^{t+\Delta t}$ is simulated by solving the mass balance equation for pollutant $p$ and accounting for in-stream decay processes following Eq. [3]. $BG_{p,n}$ represents the background concentration of pollutant $p$ in gridcell $n$. For TDS, these are estimated based on minimum observed EC-converted to TDS observations (Walton, 1989) contained in a new global salinity dataset (Thorslund and Van Vliet, 2020) and are applied as a constant background concentration. Conversely, $BG_{BOD,n}$ and $BG_{FC,n}$ are assumed to be negligible, relative to the mass of pollution produced by anthropogenic activities.

$$M_{p,n}^{t+\Delta t} = \left(M_{p,n}^{t=0} + \left(\Sigma\left(RL_{p_{n-1\to n}}\right) - RL_{p_{n\to n+1}} + \frac{L_{p_n}}{D}\right)\Delta t\right) \cdot e^{-k_{p,n}\left(\frac{\Delta t}{D}\right)}$$

[3]

Where, at the subsequent timestep interval $(t + \Delta t)$, each gridcell $n$ contains the mass of pollutant $p$ from the previous timestep $(M_{p,n}^{t=0})$ plus the pollutant load (mass second$^{-1}$) that has been transported from the immediately (adjacent) upstream gridcell(s) $(RL_{p_{n-1\to n}})$ and minus the pollutant load (mass s$^{-1}$) that has been transported downstream $(RL_{p_{n\to n+1}})$ in the time interval $\Delta t$ (s). $L_{p,n}$ represents the daily influx of pollutant loadings produced into gridcell $n$ (mass day$^{-1}$), which are added to the stream in equal increments per sub-daily timestep $\Delta t$ (s) relative to the total length of a day $D$ in seconds (i.e. 86,400 s day$^{-1}$). Our approach for adding local pollutant loadings in equal increments per sub-daily timestep is necessary as we lack information regarding the (sub-diurnal) timing at which pollution enters the stream network.

$k_{p,n}$ represents a pollutant $p$ and gridcell $n$ specific decay rate (day$^{-1}$). While we model TDS as a conservative substance (i.e. $k_{TDS,n} = 0$), we determine the first-order degradation rate of BOD $(k_{BOD_n})$ as a function of water temperature (Eq. [4]) and of FC $(k_{FC_n})$ as function of water temperature, solar radiation and sedimentation (Eq. [5]). Decay is implemented directly into DynQual by assuming decay to occur at an equal rate over the course of a day $\left(\frac{\Delta t}{D}\right)$. This assumption is necessary because we do not have sub-daily input data for some terms of the decay equations, such as water temperature (Tw) and incoming solar radiation ($I_o$).

$$k_{BOD,n} = k(20) \cdot \Theta^{(Tw_n - 20)}$$

[4]

Where $k(20)$ is a first-order degradation rate coefficient at 20°C (day$^{-1}$) assumed at 0.35 (Van Vliet et al., 2021), $Tw_n$ is the water temperature (°C) in gridcell $n$ and $\Theta$ is a temperature correction assumed to be 1.047 as per previous assessments (Wen et al., 2017; Van Vliet et al., 2021).

$$k_{FC_n} = k_d\Theta^{(Tw_n - 20)} + k_s\frac{I_o}{k_e H}\left(1 - e^{-k_e H}\right) + \frac{v}{H}$$

[5]

Where $k_d$ is dark inactivation (day$^{-1}$), $\Theta$ is a temperature correction, $Tw_n$ is the water temperature (°C) in gridcell $n$, $k_s$ is sunlight inactivation (m$^2$ W$^{-1}$), $I_o$ is the surface solar radiation (W m$^{-2}$), $k_e$ is an attenuation coefficient (m$^{-1}$), $H$ is stream depth (m) and $v$ is the settling velocity (m day$^{-1}$). Parameter values (Table 1) and mean basin average total suspended solids (Beusen et al., 2005) are

based off previous fecal coliform modelling studies (Reder et al., 2015). Parameter values, including decay coefficients, can alternatively be defined by the user directly in the source code.

**Table 1.** Assumed parameter values for fecal coliform modelling

| Variable | Unit | Value |
|----------|------|-------|
| $k_d$ | day$^{-1}$ | 0.82 |
| $\Theta$ | - | 1.07 |
| $k_s$ | m$^2$ W$^{-1}$ | 0.0068 |
| $k_e$ | m$^{-1}$ | 0.0931TSS + 0.881 |
| $v$ | m day$^{-1}$ | 1.656 |


### 2.3 Pollutant loadings

In both model configurations (stand-alone or coupled to PCR-GLOBWB2), user-defined pollutant loadings can be directly imposed on the model (akin to a forcing). Users can estimate pollutant

loadings using their preferred methodology, and subsequently route these through the global stream network, account for in-stream decay processes and calculate in-stream pollutant concentrations using the DynQual model framework. Pollutant loadings that are prescribed to DynQual directly should have a daily temporal resolution (e.g. g day$^{-1}$ or cfu day$^{-1}$).

Alternatively, when running DynQual coupled with PCR-GLOBWB2, pollutant loadings (with a daily

temporal resolution) can be simulated within the model runs, requiring only simple input data (Figure 1 and SI Section 1). This option is beneficial for users who do not have pre-calculated pollutant loadings. Furthermore, this option may be useful for those interested in scenario modelling, as input files related to different scenarios can be altered to reflect alternative climate and socioeconomic conditions.

In this set-up, DynQual estimates and routes pollutant loadings individually and combined for the main water use sectors (domestic, manufacturing, livestock and irrigation) and from urban surface runoff at 5 arc-minute spatial resolution. Loadings from the domestic sector are estimated by multiplying the gridded population with region-specific per capita excretion rates (SI Section 1.1, Table S1). For the manufacturing sector, a mean effluent concentration is multiplied by location-

specific gridded estimates of return flows from the manufacturing sector (SI Section 1.2, Table S2). Urban surface return flows are approximated by multiplying surface runoff (simulated by PCR-GLOBWB2) with the gridded urban fraction, which are multiplied by a region-specific mean urban surface runoff effluent concentration (SI Section 1.3; Table S3). The livestock sector is sub-divided into 'intensive' and 'extensive' production systems based on livestock densities to better account for

differences in the paths by which waste enters the stream network (SI Section 1.4, Table S4). Gridded livestock numbers for buffalo, chickens, cows, ducks, goats, horses, pigs and sheep are multiplied by pollutant excretion rates per livestock type and by region (SI Section 1.4, Table S5 – S7). TDS loadings from the irrigation sector are estimated by multiplying irrigation return flows simulated by PCR-GLOBWB2 with spatially-explicit mean irrigation drainage concentrations based on salinity (as

indicated by electrical conductivity) over the top- and sub-soil (SI Section 1.5). Thermal effluents (heat dumps) from thermoelectric powerplants are included as point sources of advected heat by considering the temperature difference between the flows and ambient surface water temperature conditions (SI Section 1.6). Pollutant loadings from the domestic, manufacturing and intensive livestock sectors, and from urban surface runoff, are abated based on gridcell-specific wastewater

practices. The proportion of pollutant loadings removed by wastewater treatment practices is estimated by multiplying the fraction of each treatment level occurring in a gridcell by the pollutant

removal efficiency associated with that treatment level, as described in detail in previous work (Jones et al., 2021; Jones et al., 2022).

A detailed explanation of how pollutant loadings are estimated within DynQual is provided in SI Section 1, including equations (SI Eqs. [1-8]), data sources and all parameter estimates (SI Table S1-S7) .

## 3.    Model demonstration

### 325    3.1 Model run setup

DynQual is run for the time period 1980 – 2019 using W5E5 forcing data (Cucchi et al., 2020; Stefan et al., 2021) in the configuration coupled with PCR-GLOBWB2. We used the standard parameterisation of PCR-GLOBWB2 for hydrological simulations, as described in previous work (Sutanudjaja et al., 2018). The focus of our model demonstration is on TDS, BOD and FC, as results

for Tw have been displayed extensively in previous work (Wanders et al., 2019). Pollutant loadings of TDS, BOD and FC are estimated within the model run at the daily timestep using input data summarised in Table 2 and as detailed in Section 2.3 and SI Section 1. Both the meteorological forcing data and input data used for simulating pollutant loadings used in this study are accessible through links provided. We also provide the model code and full input data required for running an

example catchment (Rhine basin) in the code and data availability statement.

**Table 2.** Summary of key input data used for the estimation of pollutant loadings in the presented model application

| Sector | Data | Source | Spatio-temporal resolution |
|---|---|---|---|
| Domestic | Population | (Lange and Geiger, 2020) | 5 arc-min; annual |
| | Excretion rates | (UNEP, 2016; Van Vliet et al., 2021) | Regional; constant |
| Manufacturing | Manufacturing return flows | PCR-GLOBWB2 (*simulated*) | 5 arc-min; daily |
| | Effluent concentrations | (UNEP, 2016; Van Vliet et al., 2021) | Global; constant |
| Urban surface runoff | Urban surface runoff | PCR-GLOBWB2 (*simulated*) | 5 arc-min; daily |
| | Effluent concentrations | (UNEP, 2016) | Regional; constant |
| Livestock | Livestock populations | (Gilbert et al., 2018) | 5 arc-min; annual |
| | Excretion rates | (Weaver et al., 2005; Wilcock, 2006; Robinson et al., 2011; Wen et al., 2017; Vigiak et al., 2019; Van Vliet et al., 2021) | Regional; constant |
| Irrigation | Irrigation return flows | PCR-GLOBWB2 (*simulated*) | 5 arc-min; daily |
| | Effluent concentrations | (Batjes, 2005) | 30 arc-min; constant |
| Power | Power return flows | (Lohrmann et al., 2019) | 5 arc-min; annual |
| | ΔT | (Van Vliet et al., 2012a) | Global; constant |

As per PCR-GLOBWB2 (Sutanudjaja et al., 2018), in addition to the original water temperature model DynWat (Wanders et al., 2019), no calibration was performed. The process-based nature and global scale of DynQual, combined with strong spatial biases in observations (Figure S2) and the large number of parameters that need to be estimated, complicate meaningful calibration. In addition, uncalibrated physical models can theoretically be applied in ungauged basins without loss of performance and are more preferable for global change assessments with different climatic and socio-economic scenarios (Hrachowitz et al., 2013; Wanders et al., 2019).

## 3.2 Model evaluation

Model simulations were compared to observations from surface water quality monitoring stations worldwide at daily temporal resolution. Observed data was obtained from various state-of-the-art databases (SI Section 3.1). Water quality monitoring data covers the entire modelled time period (1980 – 2019) and includes a far greater number of observations than in previous surface water quality modelling validation procedures (Table S8). However, monitoring stations are unevenly distributed across space, with a strong bias towards North America and Western Europe for all water quality constituents (Figure S2). Furthermore, observations at monitoring stations are highly fragmented across time, particularly for BOD and FC (Figure S2).

The overarching purpose and applications of a model, including large-scale water quality models (Beusen et al., 2015; UNEP, 2016), must be considered both for determining suitable metrics for model evaluation and for judging model performance. Given the approximations in the model, uncertainties in input data and the overall complexity in the drivers of pollutant loadings, the purpose of global water quality models is not to compute daily concentrations exactly (UNEP, 2016). The modelling strategy is thus to focus on the main spatial and temporal drivers of pollution in river networks globally to facilitate first-order approximations of in-stream concentrations. A key reason for implementing DynQual at 5 arc-minute spatial resolution is due to the marked improvement of the performance of both PCR-GLOBWB2 (e.g. discharge) (Sutanudjaja et al., 2018) and DynWat (e.g. water temperature) (Wanders et al., 2019) at finer spatial extents. These two factors have an important influence on simulated in-stream concentrations due to dilution and in-stream decay processes, respectively.

Given these factors, combined with limitations in the observational records of surface water quality (SI Section 3.1), global water quality models have typically not been evaluated with metrics commonly used for hydrological modelling such as coefficients of determination, Nash-Sutcliffe efficiency (NSE) and Kling-Gupta efficiency (KGE) (Voß et al., 2012; Beusen et al., 2015; UNEP, 2016; Wen et al., 2017; Van Vliet et al., 2021), with the exception of water temperature simulations (Van Vliet et al., 2012b; Wanders et al., 2019). The model evaluation approach adopted for DynQual combines methods applied for the evaluation of other global water quality modelling efforts. Simulated TDS, BOD and FC concentrations are evaluated with respect to pollutant classes linked to key sectoral water quality thresholds (UNEP, 2016; Wen et al., 2017) (SI Section 3.1.2; Table S9) and statistically using normalised root mean square error (nRMSE) (Beusen et al., 2015; Van Vliet et al., 2021) (SI Section 3.2.2; SI Eq. [11]). This provides an indication of prediction errors across the different water quality constituents comparable with previous large-scale water quality assessments. Conversely, the quality of water temperature simulations is evaluated using KGE (SI Section 3.2.2; SI Eq. [10]). All four water quality constituents are also evaluated by considering long-term time-series and multi-year annual cycles at individual monitoring stations (SI Section 3.2.3), which we present for the station with the most data availability across all four constituents (see Figure 5 for a station in the Mattaponi river in the USA) and for a selection of additional monitoring stations per water quality constituent (Figures S5- S8).

Overall, a strong correspondence between simulated and observed concentrations classes is found, indicating that the model is (largely) able to simulate concentrations within the correct concentration range (Figure 3). The simulated concentration class matches the observed concentration class exactly in 69%, 51% and 44% of instances for TDS, BOD and FC, respectively. When considering ± 1 pollutant class, these percentages rise to 92%, 79% and 79%. Of the mismatches in simulated and observed concentration classes, DynQual tends to underestimate TDS and BOD concentrations relative to observed in-stream concentrations (i.e. difference in classification level >=1). This occurs for 75% of mismatches in simulated TDS classes and 69% of mismatches in BOD classes. Conversely, FC mismatches occur both for under-estimates (57% of cases) and over-estimates (43% of cases) in more equal proportions.

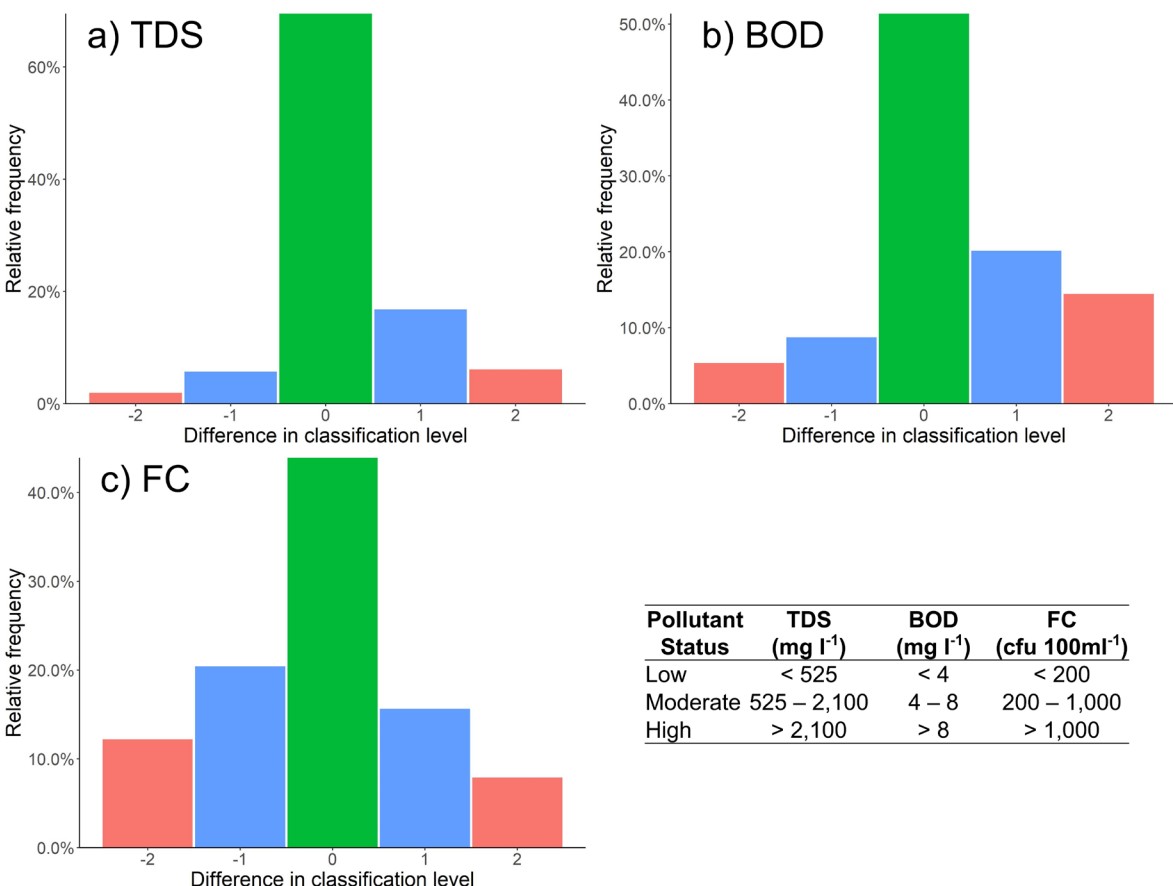

**Figure 3.** Differences in observed vs simulated pollutant classes for a) total dissolved solids (TDS), b) biological oxygen demand (BOD) and c) fecal coliform (FC). Pollutant classes are defined based on water use and ecological limitations, as stated by governmental and international organisations. A difference in classification level of "0" indicates the simulated pollutant class matches the observed pollutant class, while negative differences indicate that observed concentrations exceeded simulated concentrations, and vice-versa for positive differences.

Statistical evaluation of the water temperature simulations using the KGE coefficient demonstrates the strong performance of DynQual (Figure 4a) across all world regions (Figure S3). Across all observation stations, a median KGE of 0.72 is found (25th percentile = 0.52, 75th percentile = 0.83), with 32% of stations with KGE > 0.8, 83% of stations with KGE > 0.4 and 99% of stations with KGE values exceeding the performance threshold of > -0.41 (Knoben et al., 2019). Detailed time-series of individual rivers also demonstrate the ability of DynQual to closely replicate observed water

temperature at the daily timestep, in addition to seasonal patterns, across different world regions (Figure 5, Figure S5). A detailed evaluation of water temperature simulations is available in previous work (Wanders et al., 2019).

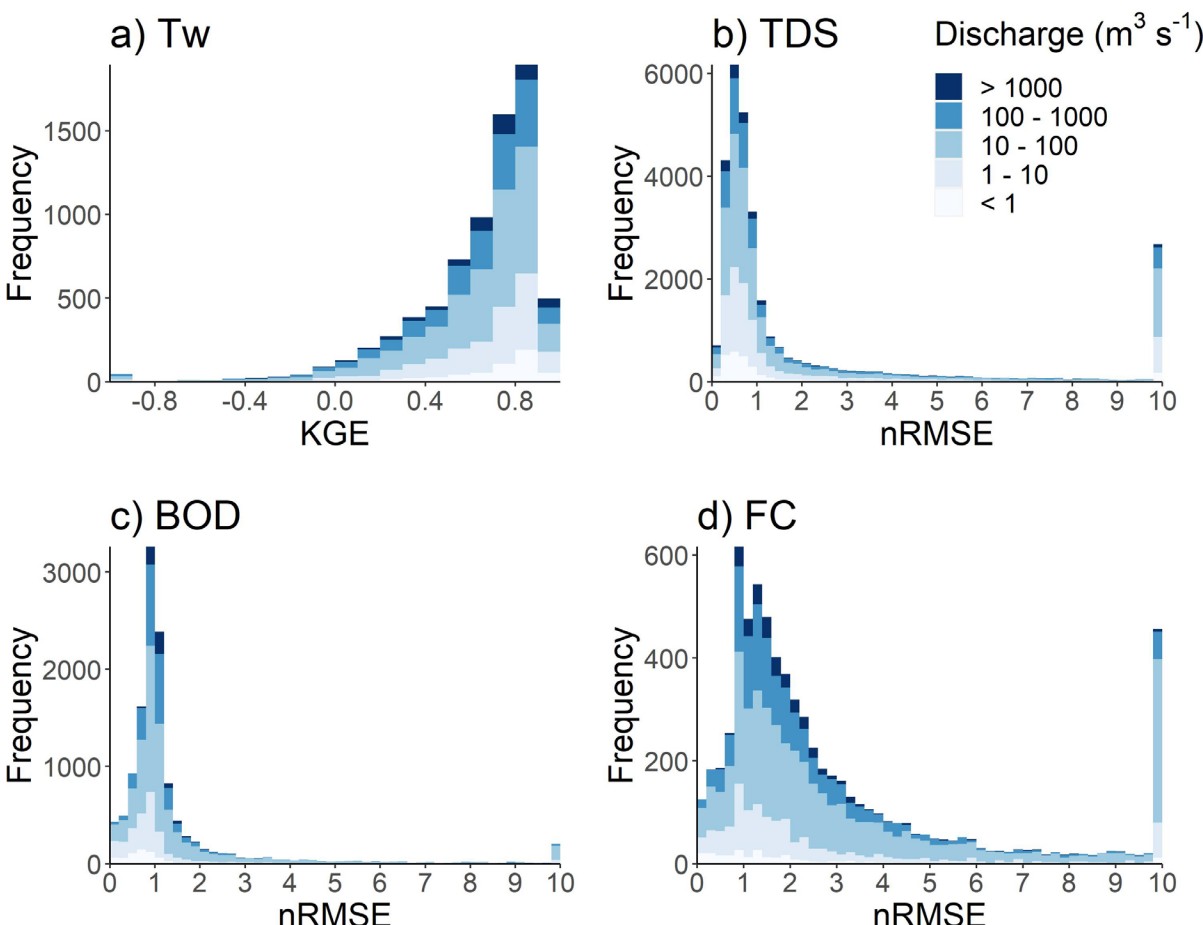

**Figure 4.** Evaluation of model performance using the Kling-Gupta efficiency (KGE) coefficient for a) water temperature (Tw); and normalised root mean square error (nRMSE) for b) total dissolved solids (TDS), c) biological oxygen demand (BOD) and d) fecal coliform (FC) simulations. Spatial patterns in KGE for Tw (Figure S3) and nRMSE for TDS, BOD and FC (Figure S4) are displayed in SI Section 3.2.2.

The distribution of nRMSE values, sub-divided by annual average river discharge, for TDS, BOD and FC is displayed in Figure 4b-d. Statistical evaluation of the simulations using nRMSE shows mixed results. A median nRMSE value of 0.76 is found for TDS across all observation stations, with 25th percentile of 0.79 and a 75th percentile of 1.83 (Figure 4b). For BOD simulations, a median nRMSE of 0.98, 25th percentile of 0.76 and 75th percentile of 1.25 is found (Figure 4c). A large spread is
found for nRMSE values for FC simulations, with a median of 1.89, a 25th percentile of 1.16 and a 75th percentile of 3.53. Simulated TDS concentrations are typically lower than observations in many locations that are proximate to the coastline, presumably due to a combination of background TDS concentrations based upon minimum observations (and applied constantly) and DynQual not accounting for the influence of saltwater intrusion. This may somewhat explain the long tail (nRMSE
> 10) in the histogram for TDS (Figure 4b) and the disproportionate tendency of DynQual to simulate TDS concentrations that are lower than observed concentrations (Figure 3). Overall, no strong spatial patterns are found in the distribution of nRMSE values of BOD (Figure S4b) and FC (Figure S4c).

For these water quality constituents, model simulations tend to represent the observed data better in larger streams (>100 m³ s⁻¹). This is likely due to the influence of spatial mismatches between monitoring station locations and model simulations being especially important in smaller streams, where concentrations are more sensitive to natural dilution capacity (i.e. water availability) and variabilities in pollutant source contributions.

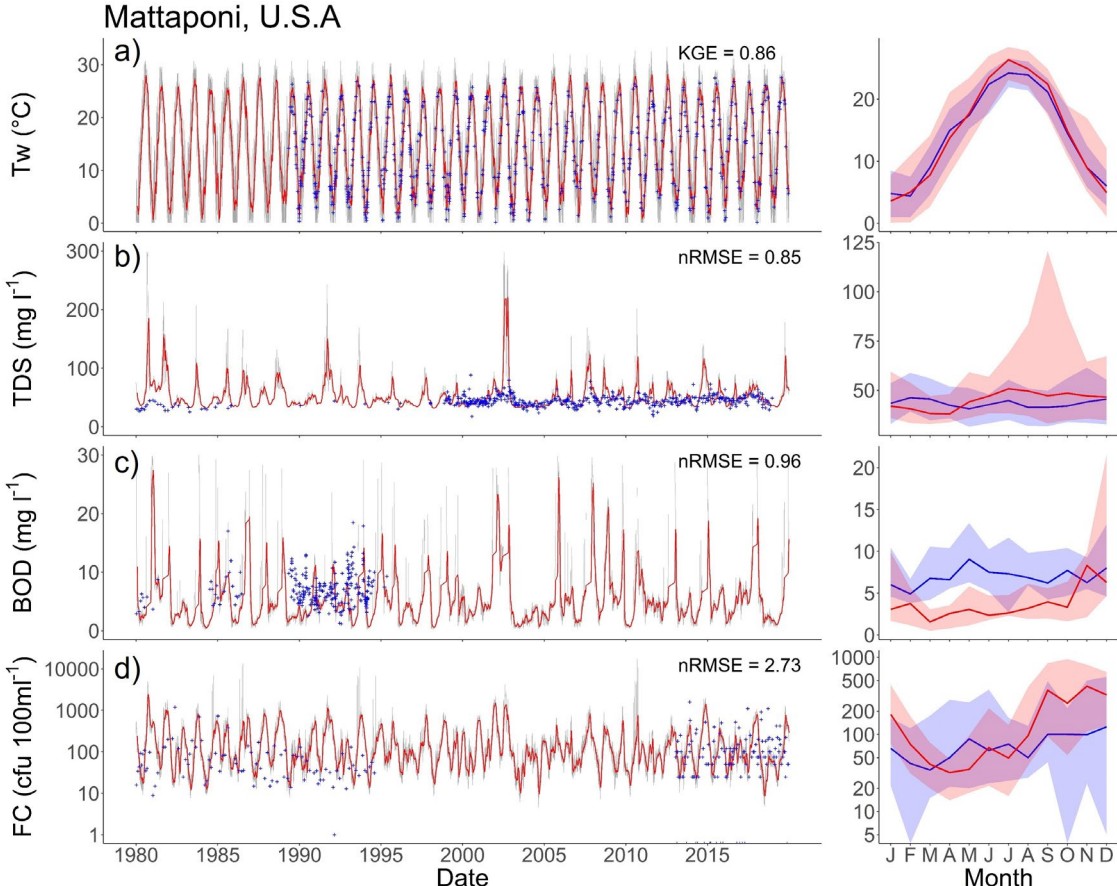

**Figure 5.** Time-series (left) and average annual cycles (right) of observed versus simulated surface water quality as indicated by a) water temperature (Tw; °C); b) total dissolved solids (TDS; mg l⁻¹) concentrations; c) biological oxygen demand (BOD; mg l⁻¹) concentrations; and d) fecal coliform (FC; cfu 100ml⁻¹) concentrations at an example water quality monitoring station. In the time-series plots, observations are indicated by blue crosses, daily simulations by grey lines and 30 day running averages by red lines. In the average annual cycles plots, blue and red lines indicated the median observed and simulated values, respectively, while the shading represents the range in values as indicated by the 10th and 90th percentiles. More examples for Tw (Figure S5), TDS (Figure S6), BOD (Figure S7) and FC (Figure S8) across different world regions are displayed in SI Section 3.2.3.

Long-term time-series and average annual cycle plots for TDS (Figure 5b; Figure S6), BOD (Figure 5c; Figure S7) and FC (Figure 5d; Figure S8) show that DynQual can generally simulate in-stream concentrations within the correct range (e.g. min-max daily concentrations, 10ᵗʰ and 90ᵗʰ percentile average annual cycles). Simulated concentrations at the example monitoring station (Figure 5) display that TDS, BOD and FC concentrations are largely simulated within plausible limits with strong overlaps in the average annual cycles, but the exact correspondence between observed and simulated concentrations at the daily timestep is relatively poor. For this observation station, simulated peaks in daily TDS, BOD and FC concentrations tend to exceed those in the observational record. However, given the incomplete nature of the observed records, it is problematic to draw conclusions on whether these concentrations are plausible but unrecorded, or if DynQual is simulating unrealistic peak concentrations. For example, while DynQual captures some of the peaks in observed daily BOD

concentrations, simulated BOD concentrations exceed those in the observational record while simultaneously under-predicting average annual cycles in BOD concentrations (Figure 5). This pattern is also observable in TDS concentrations in the Mersey River (Figure S6) and FC concentrations in the Exe River (Figure S8).

While strong seasonality is present in the Tw observations, which is also well-captured by DynQual (Figure 5a; Figure S5), and in TDS concentrations to a lesser extent (e.g. Mersey & Komati rivers in Figure S6), there is an overall lack of strong seasonal patterns in the observed records of BOD and FC concentrations. This, combined with large variability in the observed concentrations, results in large uncertainty in average annual cycles of observed concentrations across all months, as indicated by 10th and 90th percentiles (Figure 5c-d; Figure S7 – S8). Annual average cycles in observed and simulated concentrations tend to strongly overlap for both BOD and FC. However, seasonal patterns are more evident in BOD simulations than observations (e.g. Mersey, Periyar in Figure S7) and the large variability in observed FC concentrations is not replicated by DynQual daily simulations (e.g. Cauvery, Rhine in Figure S8). In the case of FC concentrations, for example, this could suggest that DynQual misses or under-represents the importance of pulse disturbances (e.g. high rainfall events causing sewer overflows) on the transport of pollutants to surface waters.

### 3.3 Spatial patterns

The spatial patterns in TDS (Figure 6), BOD (Figure 7) and FC (Figure 8) concentrations show substantial variations both within and across world regions, driven by different sectoral activities (Figure 9). The dilution capacity of rivers is also a major determinant of in-stream concentrations. Averaged at the annual time-scale this is particularly evident for BOD and FC where the large dilution capacity of some major rivers is sufficient to dilute concentrations to relatively low levels, despite often being fed by more polluted tributaries. However, it should also be noted that both river discharges and in-stream concentrations can exhibit substantial intra-annual variability, thus pollutant hotspots and the magnitude of pollutant levels must also be considered at finer temporal scales than presented here. Intra-annual variability can occur in the model due temporal variations in: 1) pollutant loadings; 2) water availability (i.e. dilution capacity) and 3) in-stream decay processes.

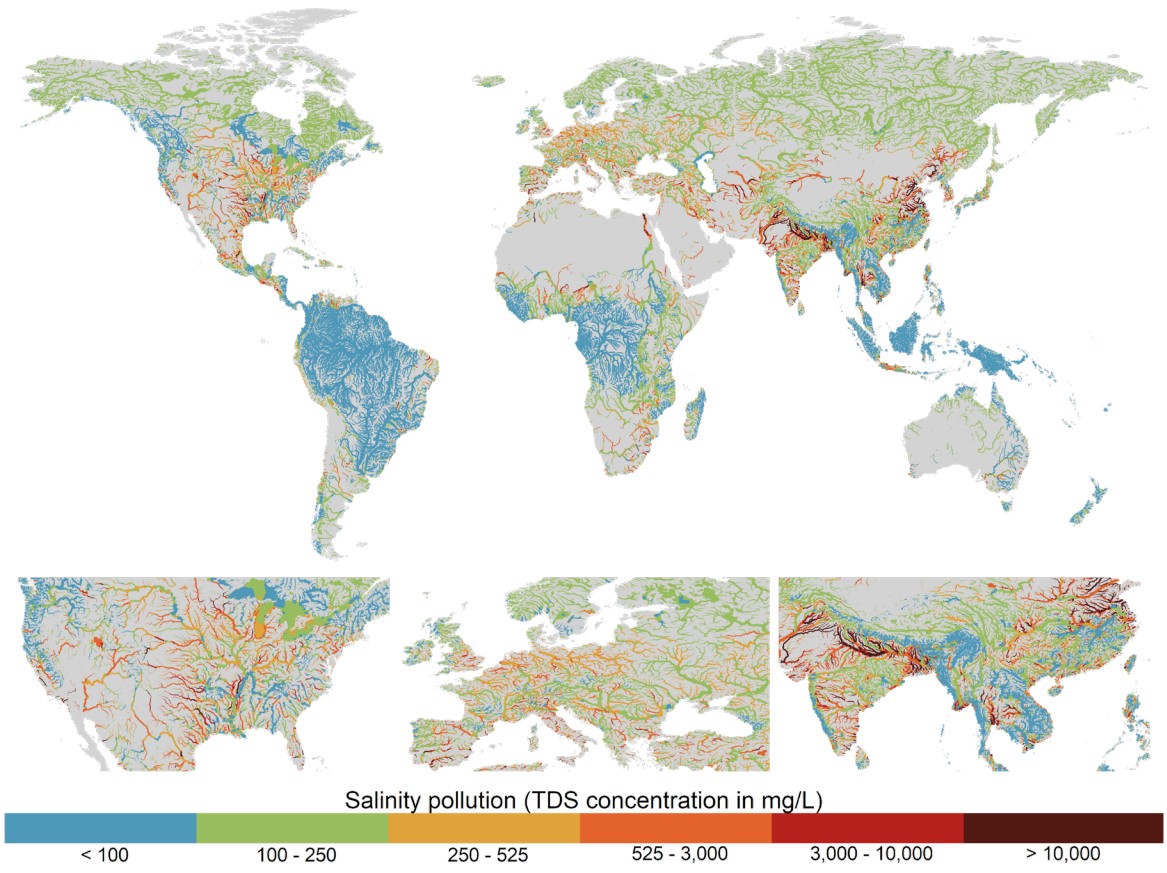

**Figure 6.** Annual average total dissolved solids (TDS) concentrations for the period 2010 – 2019. Plotted for rivers with > 10 m$^3$ s$^{-1}$ annual average discharge.

TDS concentrations show strongly regional patterns, with key hotspots of salinity pollution located in southern Asia (Pakistan and northern India) and eastern China, and to a lesser degree across the United States and Europe (Figure 6). High TDS concentrations in south-east Asia are predominantly driven by the irrigation sector and the presence of saline soils (Figure 9a). While the irrigation sector is also an important driver of TDS pollution in eastern China, the contribution from manufacturing

activities is also substantial (Figure 9a). The manufacturing sector is the dominant contributor of TDS pollution across most of North America and Western Europe, accounting for >75% of in-stream pollutant loadings in almost all major river segments in these regions (Figure 9a). Aside from the lower Nile, where salinity pollution is predominantly from the manufacturing sector, the domestic sector is the key source of (non-natural) TDS loadings in Africa. However it should be noted that,

aside from in the lower Nile, TDS concentrations are simulated to be relatively low across most of Africa (Figure 6;).

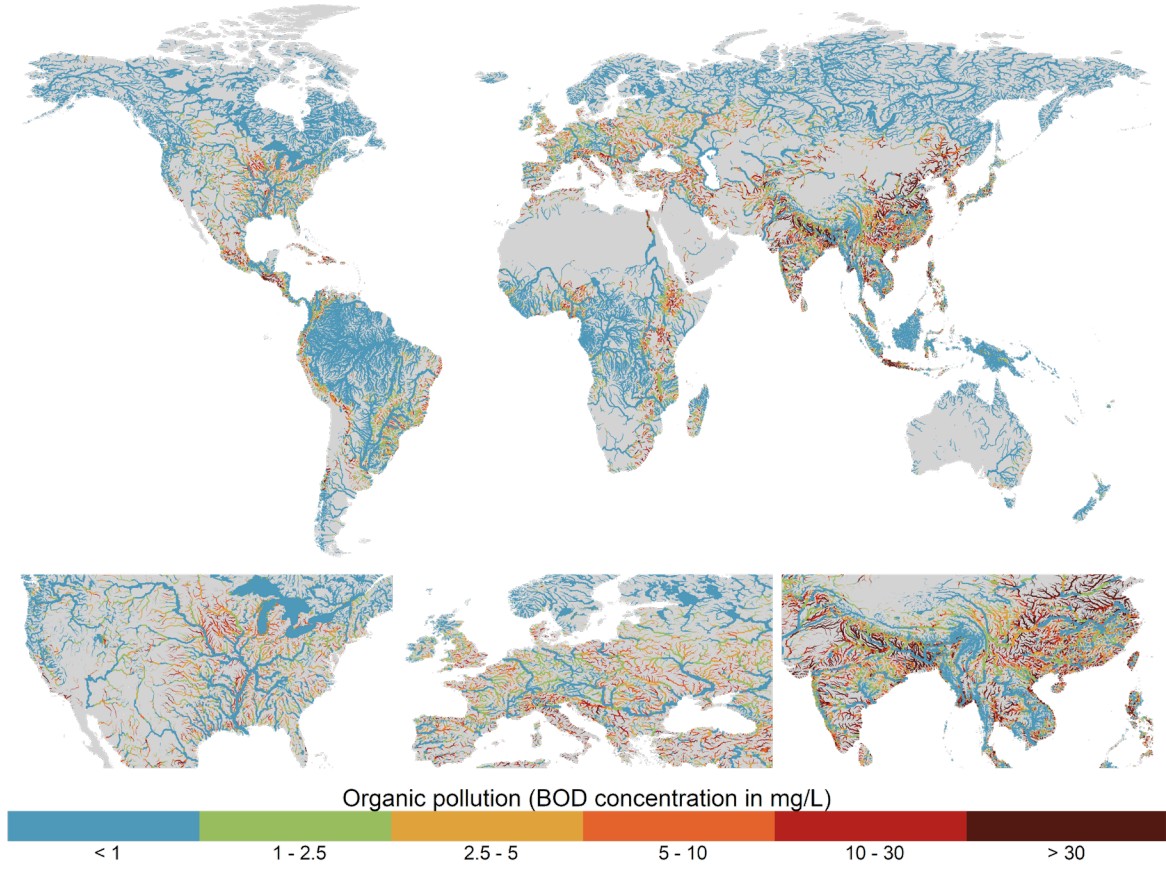

Organic pollution (BOD concentration in mg/L)

| < 1 | 1 - 2.5 | 2.5 - 5 | 5 - 10 | 10 - 30 | > 30 |

**Figure 7.** Annual average biological oxygen demand (BOD) concentrations for the period 2010 – 2019. Plotted for rivers with > 10 m$^3$ s$^{-1}$ annual average discharge.

While BOD concentrations show considerable diversity across the major world regions, a substantial proportion of river segments across populated areas of all continents experience moderate-to-high organic pollution (Figure 7). There are clear spatial patterns in the dominant sectoral activities contributing BOD loadings worldwide, and it also evident that BOD pollution in most world regions is driven by a combination of multiple sectors opposed to from an individual dominant activity (Figure 9b). Across Europe in particular, which sector is dominant varies both spatially and temporally and the contribution from the dominant sector is typically <50% (Figure 9b). The manufacturing sector is the most significant source of BOD pollution across rivers in the United States, however the relative contribution commonly falls in the 20 – 50% or 50 – 75% categories (Figure 9b). In the most polluted world regions, south and south-east Asia, typically the domestic sector is dominant. However, there are also significant contributions from manufacturing and extensive livestock activities (Figure 7; Figure 9b). Lastly, while its influence is highly localized, urban surface runoff can also represent an important source of BOD pollution in heavily urbanised gridcells across all world regions.

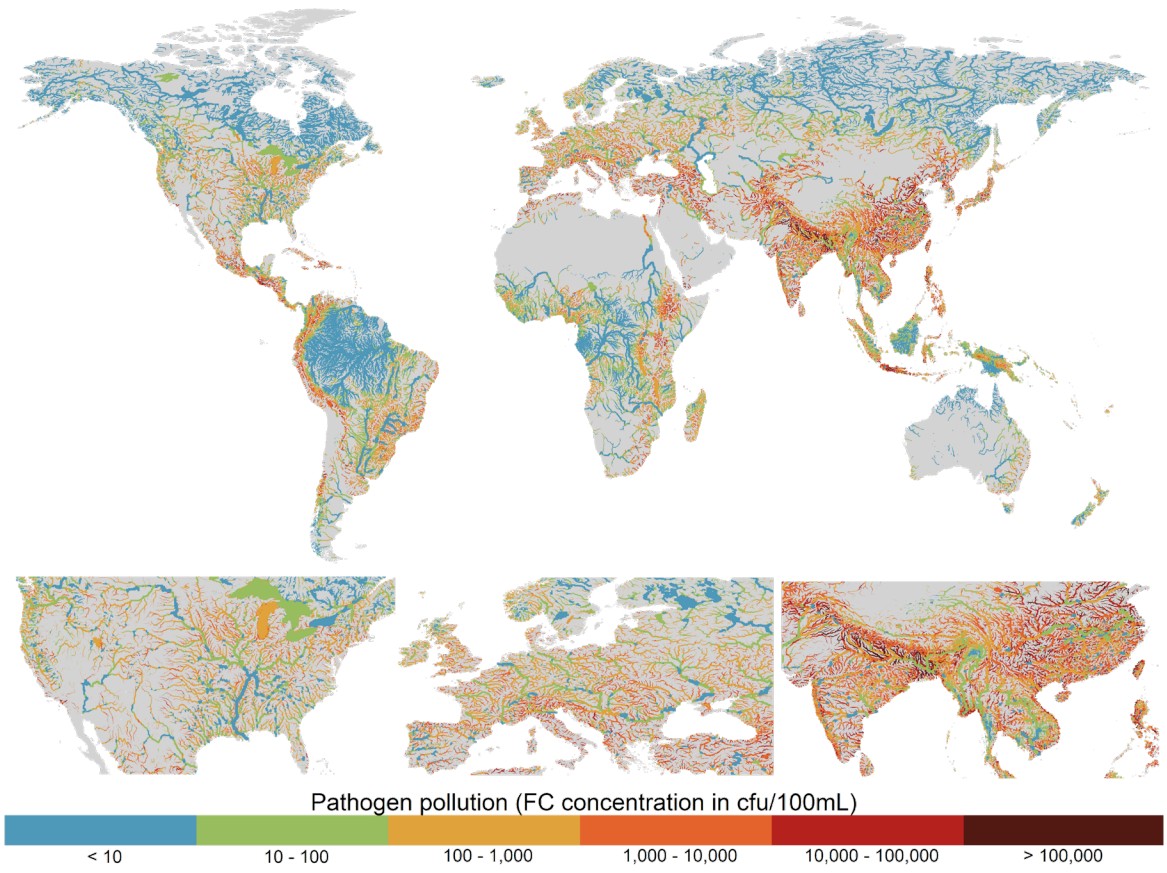

Pathogen pollution (FC concentration in cfu/100mL)

| < 10 | 10 - 100 | 100 - 1,000 | 1,000 - 10,000 | 10,000 - 100,000 | > 100,000 |

**Figure 8.** Annual average fecal coliform (FC) concentrations for the period 2010 – 2019. Plotted for rivers with > 10 m$^3$ s$^{-1}$ annual average discharge.


FC pollution is particularly high across south and south-east Asia, with more localised hotspots found in parts of western Latin America, southern Europe, Middle East and eastern Africa (Figure 8). Similar to BOD pollution, a large proportion of stream segments in south and south-east Asia are heavily polluted, with typically only rivers with extremely high dilution capacities appearing in the
lower concentration classes. In this region, the domestic sector is predominantly responsible for FC pollution (commonly > 75%), attributed to large urban populations coupled with a large proportion of domestic wastewater being inadequately treated (Figure 9c). In countries with high municipal wastewater collection and treatment rates, such as in Europe, the relative influence of livestock activities tends to be larger. While manufacturing activities remain the dominant source of FC
pollution in North America, despite relatively high wastewater treatment rates, the percentage contribution is typically <50% and livestock activities also represent an important source of FC loadings (Figure 9c). Despite variable municipal wastewater collection and treatment rates across Latin America, livestock activities appear to dominate FC loadings outside of the Amazon basin (Figure 9c). This can be attributed to very high livestock numbers (particularly cattle), combined with
the fact that the most of the large urban settlements (and thus domestic FC pollutant loadings) in South America are located in the coastal zone. As such, pollution from the domestic and manufacturing sectors typically enter the river network at downstream locations causing localised pollution before outflow to the ocean.

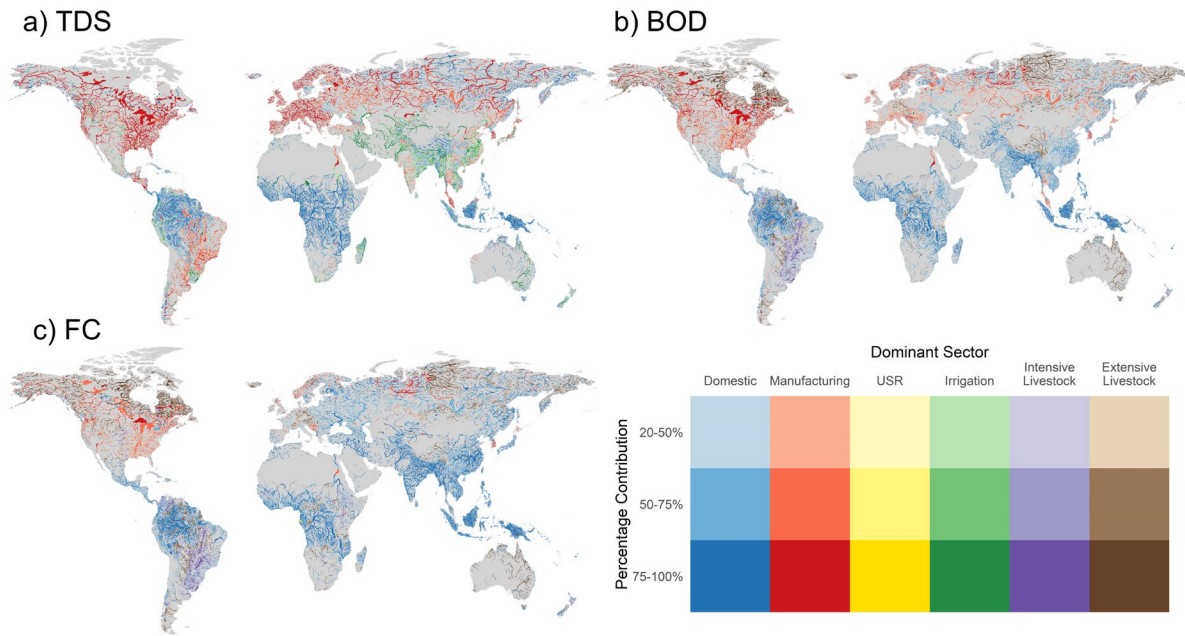


**Figure 9.** Dominant sectoral activity contributing towards a) total dissolved solids (TDS), b) biological oxygen demand (BOD) and c) fecal coliform (FC) pollution averaged over 2010 – 2019. Plotted for rivers with > 10 m$^3$ s$^{-1}$ annual average discharge.

## 3.4 Trends

Long-term trends in TDS, BOD and FC concentrations over the simulated period (1980 - 2019) are also presented (Figure 10). TDS concentrations in most world regions are either relatively constant or show relatively upwards gradual trends (Figure 10a). Typically, where TDS concentrations are increasing, the trend has been driven mainly by expansions in manufacturing or irrigation activities.
Comparatively, trends in BOD (Figure 10b) and FC (Figure 10c) concentrations are larger in magnitude and exhibit substantially more spatial variation across the major world regions. Regionally, the strongest increases in BOD and FC concentrations are found in Sub-Saharan Africa, where wastewater treatment rates are low, and south Asia, where the rate of population growth and economic development has significantly outstripped expansions in wastewater treatment infrastructure. Strong
increasing trends are also found across most of Latin America, where a significant proportion of collected wastewater does not undergo wastewater treatment (UNEP, 2016; Jones et al., 2021). BOD and FC concentrations across North American rivers have typically remained relatively constant, or exhibit small decreasing trends. Strong decreasing trends are found across Europe, including the Danube and Rhine basins. In all world regions, the influence of reservoirs on BOD and FC
concentrations is also evident, with increased water volumes (i.e. dilution) coupled with longer residence times (i.e. greater decay) reducing BOD and FC concentrations at these specific locations.

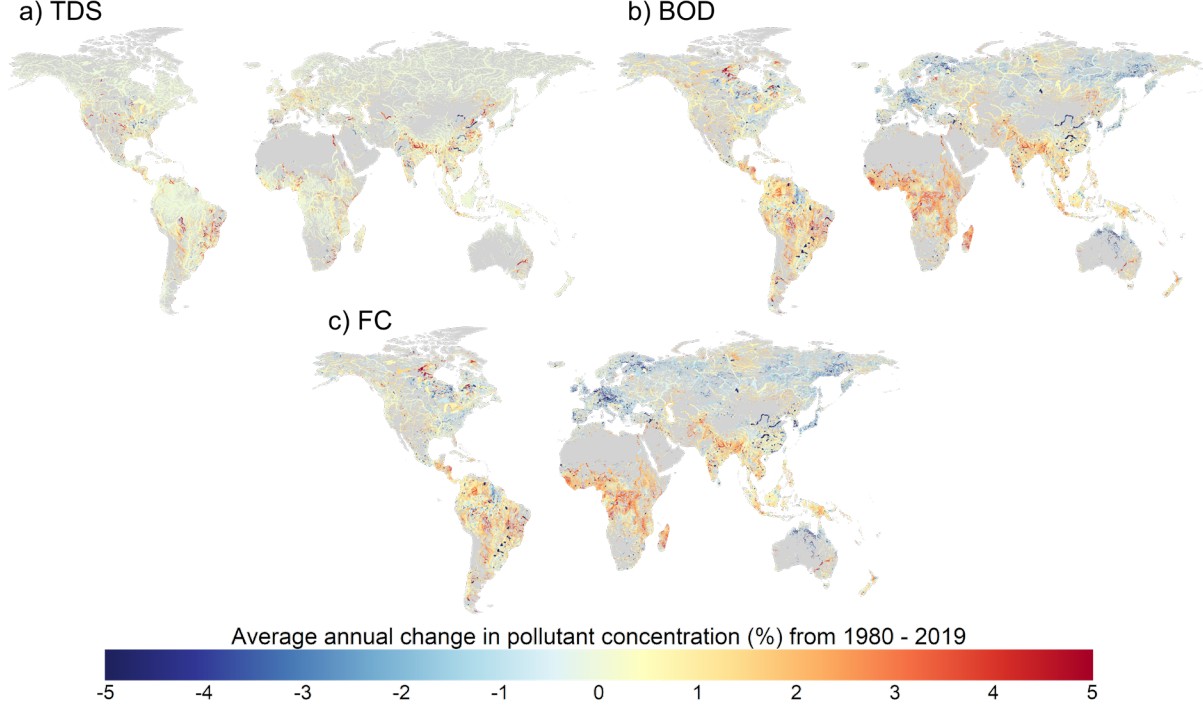

**Figure 10.** Average annual percentage changes in a) total dissolved solids (TDS), b) biological oxygen demand (BOD) and c) fecal coliform (FC) concentrations for the period 1980 – 2019. Plotted only for rivers with > 10 m³ s⁻¹ annual average discharge.

Complementary to the spatial analysis, we considered the proportion of the population that inhabits gridcells exhibiting different trends in pollutant concentrations, aggregated by geographical region and economic classification (Figure 11). It should be noted that trends (Figures 10 and 11) are not indicative of the degree of pollution directly, and thus should also be considered with respect to in-stream concentrations (Figures 6 - 8). Changes in TDS concentrations in the most populated areas worldwide are typically low, with increases of 0 – 1% most common across all geographical regions (Figure 11a). Conversely, strong regional patterns are evident for BOD (Figure 11b) and FC (Figure 11c) concentrations. Particularly in Sub-Saharan Africa and southern Asia, BOD and FC concentrations in populated locations have been almost exclusively increasing. Over half of the population of Sub-Saharan Africa live in areas where BOD and FC concentrations have increased (on average) by >2% per year from 1980 – 2019. Conversely, in Western Europe, trends in BOD and FC have been negative for areas where 60% of the population lives.

When aggregating trends by country-specific economic classifications, trends in TDS, BOD and FC pollutant concentrations all display a clear correlation with level of economic development (Figure 11). For the water quality constituents considered, the strongest and most widespread decreases in pollutant concentrations have been experienced by 'high-income' countries, while 'low-income' countries have experienced the greatest and most widespread degree of water quality degradation. These patterns are particularly clear for FC, where approximately 60% of the population in 'high-income' countries live in gridcells displaying negative trends in FC concentrations, compared to 50%, 25%, and 10% in 'upper-middle', 'lower-middle' and 'low-income' countries, respectively. Furthermore, in the 'low-income' countries, 50% of the population live in areas where FC concentrations have increased (on average) by 2% each year from 1980 to 2019.

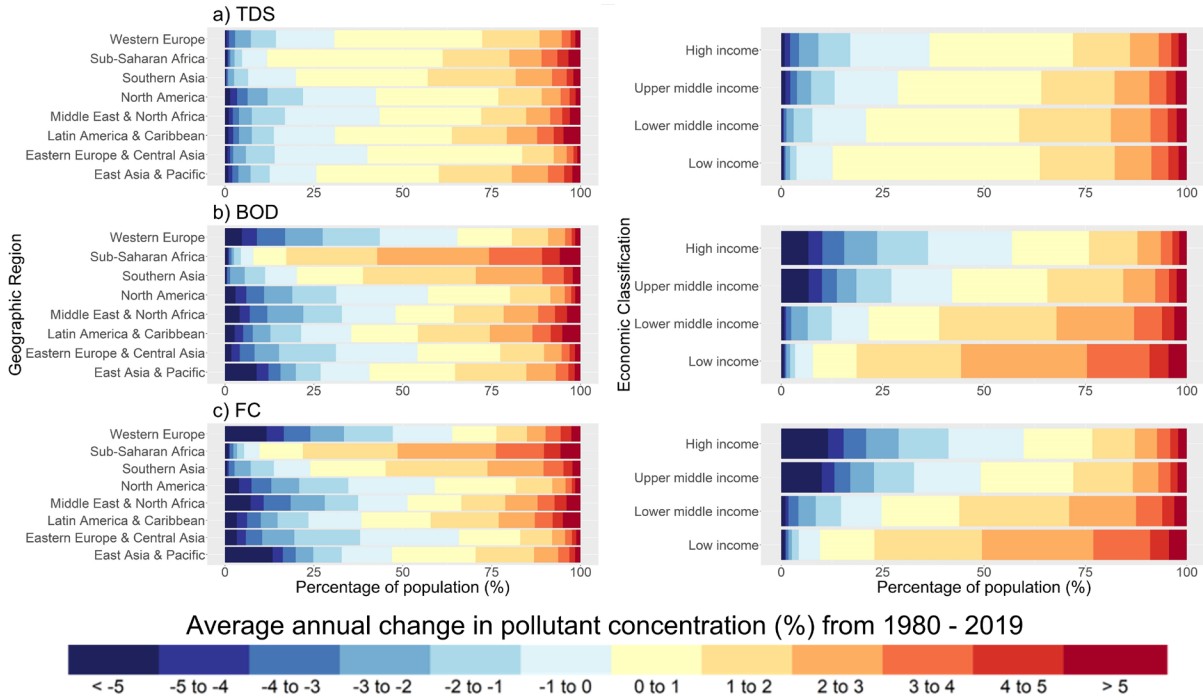

**Figure 11.** Average annual percentage changes in a) total dissolved solids (TDS), b) biological oxygen demand (BOD) and c) fecal coliform (FC) concentrations for the period 1980 – 2019. Results are displayed for the proportion of population (%) inhabiting gridcells exhibiting different trends in pollutant concentrations, aggregated by geographical region (left) and economic classification (right).

Lastly, we present time-series of in-stream TDS, BOD and FC concentrations delineated by sector-specific contributions at three selected locations (Figure 12) for which validation plots are also presented (Figures S6 – S9). While it is not our intention to explain the patterns in concentrations and sectoral drivers for the Mersey, Cauvery and Kiso rivers specifically, these plots are illustrative of the capabilities of DynQual. For example, these plots demonstrate the relative importance of different water use activities on in-stream concentrations dynamically, and also display changes over longer time periods. This is particularly evident in FC concentrations in the Mersey river, where decreasing loadings from the domestic and manufacturing sectors, primarily due to increases in wastewater treatment capacities, have driven an overall trend towards water quality improvements. Conversely, the manufacturing sector is simulated to have had an increasing influence on TDS concentrations in the Kiso river since ~2004, replacing the irrigation sector as the dominant driver of salinity pollution.

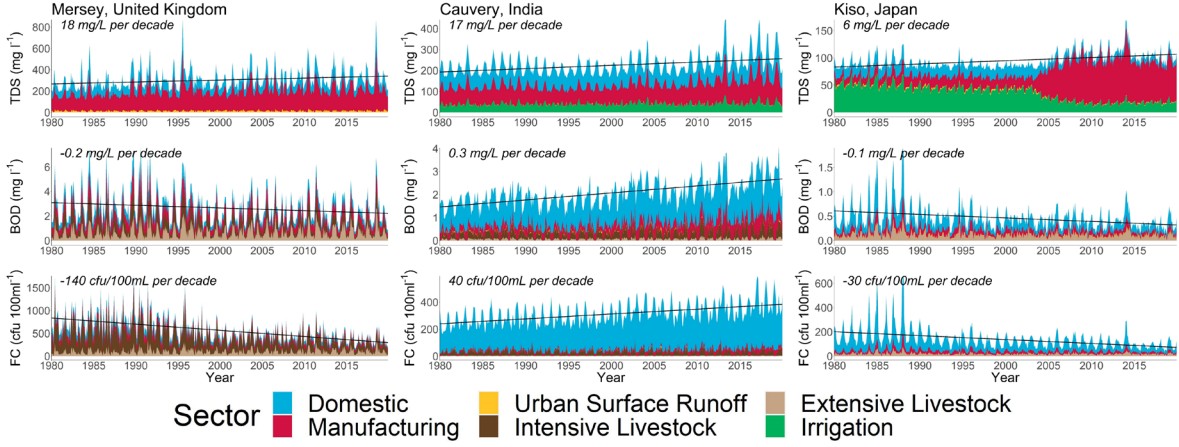

**Figure 12.** Simulated in-stream total dissolved solids (TDS; a), biological oxygen demand (BOD; b) and fecal coliform (FC; c) concentrations in selected rivers, disaggregated by contributing water use sectors and including linear decadal trends.

## 4. Discussion, conclusions and future work

To conclude, we have developed and evaluated a new global surface water quality model for simulating TDS, BOD and FC concentrations as indicators of salinity, organic and pathogen pollution, respectively. Building upon the water temperature model DynWat, and utilising approaches developed in previous water quality model efforts, the open-source code is structured in a way that allows for flexibility in both hydrological and pollutant loading inputs. Output data from DynQual has potential to inform assessments in a broad range of fields, including ecological, human health and water scarcity studies. Such work is relevant not only to the hydrological and water quality modelling communities, but also has applications for the broader scientific community in addition to informing policy regarding water resources management.

DynQual is ambitious in its aim to model global surface water quality 1) using a consistent approach; 2) dynamically; 3) considering multiple water quality constituents; and 4) at a high spatio-temporal (i.e. 5 arc-min and daily timestep) resolution. Any model must consider the trade-offs between model complexity and availability of input datasets and data to parameterise process descriptions of the model (Weaver and Zwiers, 2000; Wen et al., 2017), and the impact of this on model scope. Being a global model, DynQual is inherently unable to accurately represent all aspects relevant to the local context. Rather, the modelling strategy is to focus on the main spatial and temporal drivers of pollution in river networks globally to facilitate first-order approximations of in-stream concentrations at high spatial (5 arc-min) and temporal (daily) resolution with global coverage. As such, DynQual allows for the investigation of research questions that only large-scale modelling efforts can address. These include, as presented in the model application section, global pollution hot- and bright- spot identification (Figures 6 - 8), the relative importance of different contributing sectors to water quality status across the globe (Figure 9) and meta-trends in surface water quality dynamics (Figures 10 – 11). The dynamic nature of DynQual can also facilitate analysis of intra- and inter- annual trends in surface water quality, and help to further enhance the understanding of the main drivers of pollution via (dynamic) sectoral attribution (Figure 12). Furthermore, this approach has particular value for simulating surface water quality in ungauged catchments, and our use of globally consistent input data facilitates meaningful comparisons across different world regions. Given severe limitations in observational records of surface water quality, both in terms of spatial coverage and the number of observations per water quality monitoring station (SI Section 3.1), these are key strengths of

DynQual. However, poor data availability is simultaneously a severe limitation for both the development of global water quality models and their evaluation.

Uncertainties in surface water quality simulations arise from a combination of uncertainties associated with quantifications of pollutant loadings (e.g. pollutant excretion, emission rates and sector-specific return flows), the quality of hydrological simulations (e.g. discharge and velocities) and the representation of in-stream processes (e.g. decay coefficients). These uncertainties are especially prevalent when modelling at large spatial extents. In-stream pollutant concentrations are sensitive to
dilution capacity, thus the quality of the hydrological simulations. This issue contributes to uncertainties in simulated concentrations particularly in headwater streams. Fixed estimates of decay coefficients are assumed, which contributes to uncertainties in simulations of reactive constituents such as BOD and FC. In addition, the representation of lakes and reservoirs in DynQual is rudimentary, with total (routed) loadings instantaneously averaged over the volume of the water body
assuming full mixing.

With respect to pollutant loading quantifications, spatial mismatches between the generation of pollutant loadings and the location of entry to the stream network (return flows) can result in the simulation of unrealistic concentrations, particularly in gridcells with very low water availability (i.e. headwater streams). This can occur where the drivers of point-source pollutant emissions (e.g.
population) do not directly coincide with the location of wastewater treatment plant outlets. A lack of temporally-explicit input data can hinder proper representation of sectors with strong intra- or inter-annual variability. For instance, notable limitations for the livestock sector are the simplified assumptions made for livestock population numbers (assumed to be constant across days of the year), changes to livestock numbers across multi-year periods (applied annually and based on regional
averages) and transportation pathways to the stream network (assumed to be a function of surface runoff excluding the representation of processes that affect pollutant retention in soils). Locally relevant sources of pollution may also be entirely excluded, such as the lack of information on TDS emissions from mining activities and road-deicing. Similarly, pulses of pollutant loadings occurring during extreme rainfall of flood events are also overlooked, such as those associated with sewer
overflows or from inundated industrial areas.

Despite these uncertainties, DynQual has been demonstrated to perform with a reasonable level of performance, especially given the approximations of the model. Water temperature simulations closely match observations at daily resolution as indicated by KGE coefficients (Figure 4a), which are high across all world regions (Figure S3). Furthermore, time-series and average annual plots (Figure
5a; Figure S5) demonstrate that seasonal regimes present in observed water temperatures are well-captured by the model. Simulated TDS, BOD and FC concentrations are largely within the correct concentration classes (Figure 3) with nRMSE coefficients (Figure 4b-d) deemed reasonable considering the challenges of comparing individual (instantaneous) observed daily TDS, BOD and FC concentrations against simulated daily concentrations. Long-term time-series and average annual
cycle plots for TDS (Figure 5b; Figure S6), BOD (Figure 5c; Figure S7) and FC (Figure 5d; Figure S8) show that DynQual can generally simulate in-stream concentrations within the correct range (e.g. min-max daily concentrations, $10^{th}$ and $90^{th}$ percentile average annual cycles), but simulations of in-stream concentrations timeseries on a daily timestep show relatively poor agreement with the observed timeseries. Observed data records also tend to display large variability in concentrations but
little (systematic) seasonality, especially for BOD (Figure S7) and FC (Figure S8) concentrations. These factors have a strong influence on metrics including RMSE, but especially the other commonly-used evaluation metrics in hydrology such as the Nash-Sutcliffe efficiency (NSE) and Kling-Gupta efficiency (KGE), and hence support our decision not to evaluate model performance using these metrics. Challenges related to the observational records themselves should also be acknowledged.
These can relate to, for example, artefacts in observational records (Figure S9a), issues related to instrument detection limits and/or reporting accuracies (Figure S9b) and large variability in the

observation records (Figure S9c). Lastly, given the approximations of the model, the overall complexity in the drivers of pollutant loadings and input data limitations, we re-iterate that the current set-up of DynQual is not suited to simulate daily TDS, BOD and FC concentrations that correspond exactly with in-situ observational measurements.

With few comparable studies in the current literature, it is difficult to quantitatively assess the performance of DynQual relative to other large-scale surface water quality models. Overall, our modelled spatial patterns in surface water quality match well with previous regional and global assessments – displaying multi-pollutant hotspots (e.g. TDS, BOD, FC) to be located across northern India and eastern China in particular (UNEP, 2016; Wen et al., 2017; Van Vliet et al., 2021). Consistent with a recent data-driven (machine learning) approach (Desbureaux et al., 2022), albeit for some different water quality constituents (e.g. total phosphorus), we find a general trend towards surface water quality improvement in development countries and deterioration in developing countries. Water temperature (Tw) simulations closely match those of the global water temperature models upon which DynQual is based (Van Vliet et al., 2012b; Wanders et al., 2019; Van Vliet et al., 2021). For total dissolved solids (TDS) and biological oxygen demand (BOD) concentrations, values of and patterns in normalised root mean square errors (nRMSE) are similar to previous work (Van Vliet et al., 2021), with reasonable model performance (<1 nRMSE) exhibited at monitoring locations across all continents. Other large-scale surface water quality models have validated simulated concentrations with respect to concentration classes linked to sectoral water use and environmental health limits. Following this approach, Wen et al., (2017) reported BOD concentrations simulated within the same classification in 94% of instances, however this is based on only 760 measurements of which 91% are modelled in the lowest pollutant class (0 – 5 mg l[-1]). More comparable to our simulations, UNEP (2016) compared modelled and observed pollutant classes for TDS, BOD and fecal coliform (FC) concentrations across Latin America, Africa and Asia, achieving largely comparable model performance. Comparing our simulations to output from other global water quality models modelling Tw, BOD, TDS and FC, when available, will provide further insights into model performance.

Meaningful comparisons to other surface water quality models are challenging due to the high diversity in terms of: 1) spatial extent (e.g. lumped vs. distributed); 2) temporal resolution (e.g. daily vs. monthly vs. annual vs. decadal); and 3) water quality constituent and reporting form (e.g. loads vs. concentrations). Similarly, watershed-scale surface water quality models are constructed for different purposes than large-scale (continental to global) surface water quality models. These watershed models can better incorporate locally relevant input data and processes, are parameterized for local conditions and typically have data of good quality and record length for calibration and validation – which facilitates higher precision and accuracy in both hydrological and water quality simulations. However, these models are reliant upon detailed local knowledge which is severely lacking for many (particularly ungauged) catchments worldwide (e.g. large parts of Africa).

Despite their limitations, process-based large-scale water quality models can facilitate first-order assessments of global water quality dynamics that are consistent across both space and time, such as those demonstrated in the model application section of this study. Future applications of DynQual may include: 1) expanding the number of modelled water quality constituents; 2) further spatio-temporal analysis of surface water quality; and 3) investigating the impact of uncertain climatic and socio-economic change on future surface water quality.

## 5. Code and data availability

DynQual v1.0 is open source and distributed under the terms of the GNU General Public License version 3, or any later version, as published by the Free Software Foundation. The full model code, configuration INI files and a user manual is provided through a GitHub repository: https://githubv.com/UU-Hydro/DYNQUAL. The model code presented in this manuscript is archived at https://doi.org/10.5281/zenodo.7398410.

A full set-up with all required input datasets for running DynQual for the Rhine-Meuse basin is provided as an example (https://doi.org/10.5281/zenodo.7027242). Monthly water temperature (Tw) and salinity (TDS), organic (BOD) and pathogen (FC) concentrations are available directly via https://doi.org/10.5281/zenodo.7139222. Here, we also provide the output hydrological data (discharge and channel storage) simulated within the model run.

## 6. Author contribution

The research was designed by ERJ, MFPB and MTHvV. The surface water quality model was developed by ERJ, with assistance from NW and EHS. Output data analysis and presentation of results was led by ERJ, with guidance and feedback from MFPB, NW, LPHvB and MTHvV. All authors contributed to and approved the manuscript.

## 7. Competing interests

The authors declare no conflict of interest.

## 8. Acknowledgements

MTHvV was financially supported by a VIDI grant (Project No. VI.Vidi.193.019) of the Netherlands Scientific Organisation (NWO). NW acknowledges funding from NWO 016.Veni.181.049. We acknowledge the Netherlands Organisation for Scientific Research (NWO) for the grant that enabled us to use the national supercomputer Snellius.

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
