# Peer review of "DynQual v1.0: A high-resolution global surface water quality model"

_Geoscientific Model Development, 2022_

## Author Comment (AC1)

**Community comment: Jason Ke**

This manuscript developed a global water quality model DynQual V1.0 and interpreted its results for TDS, BOD, and FC. Overall this manuscript is well-written with good-quality figures. Model results regarding the spatial patterns of concentration and temporal trends by region and economic development are interesting. However, there are some concerns about the model evaluation.

1) there seems no description of model calibration. How was the calibration done for the global water quality model? Is it a simultaneous calibration for both hydrology (discharge) and water quality (Tw, TDS, BOD, FC), or a two-step calibration strategy with discharge calibrated first followed by water quality calibration? Since the author mentioned that discharge was very important for model results (Supplement, Line 295), I would assume the discharge has to be well-calibrated before modeling water quality.

*DynQual*, in addition to the underlying model *PCR-GLOBWB2* (Sutanudjaja et al., 2018) and the original water temperature model *DynWat* (Wanders et al., 2019), are uncalibrated. This is an important point that is currently missing, and one that we will explicitly state and justify in the manuscript during revisions.

The process (physically) based nature and global scale of our model, combined with large data gaps in both space and time complicate meaningful calibration. For both the hydrology and water quality aspects, we want to avoid the creation of 'calibration artefacts' – whereby deficiencies in process descriptions are concealed by parameter estimation and there is a tendency to be biased towards areas/subsets where data availability is high, which could introduce a lack of spatial parameter consistency between different locations. This is especially problematic for calibrating large-scale water quality models, due to the strong spatial biases in the observations and the large number of parameters that need to be estimated. On the other hand, uncalibrated physical models can theoretically be applied in ungauged basins without loss of performance (Hrachowitz et al., 2013; Wanders et al., 2019).DynQual will also be used for global change assessments with different climatic and socio-economic scenarios, for which we preferably work with uncalibrated models.

All input parameter values for the global water quality modelling work are derived from previous (global) work (e.g. UNEP 2016, Reder et al., 2015, van Vliet et al., 2019). While PCR-GLOBWB2 is also parameterised on the basis of existing global datasets without further calibration, extensive model validation of hydrological simulations (e.g. discharge) have been performed using GRDC discharge data (5,363 stations) and GRACE total water storage thickness, as described in more detail in Sutanudjaja et al., (2018).

Please also note that the majority of global water quality models are currently uncalibrated, including QUAL (van Vliet et al., 2021), WorldQual (Voß et al., 2012) and IMAGE-GNM (Beusen et al., 2015) due to their process-based nature and aim of simulating water quality in data scare regions.

2) The model evaluation that is very important to the model development paper seems underdeveloped. It is essential to evaluate the model performance before the model result interpretation. For example, it is ideal to evaluate model performance whenever data are available. For example, there are 27,238 stations with TDS data in the Supplement. Perhaps the author could do the following evaluation regarding 1) spatial pattern of mean concentration (e.g., model mean vs. data mean from the station with high data availability); 2) temporal dynamics regarding seasonal fluctuations and long-term trends (e.g., Fig 11, add data points to the temporal trend plots to evaluate if the model could reproduce the long-term trends)

We agree that model evaluation is very important, and is somewhat underdeveloped in the current submission. Please note that some validation results are also presented in the Supplementary Information of Jones et al., (2022) (https://www.nature.com/articles/s43247-022-00554-y). We will endeavor to further improve this section, both in the manuscript and SI, e.g. by adding global maps of time-average model evaluation statistics at locations with water quality observations.

Overall, and as argued in other global water quality modelling work (e.g. Beusen et al., 2015; UNEP 2016) we believe that the overarching purpose and applications of the model must be considered when evaluating the performance. DynQual is ambitious in its aims to model surface water quality 1) using a consistent approach at the global scale; 2) dynamically; 3) considering multiple water quality constituents (multipollutant approach); and 4) at a high spatiotemporal resolution. We believe the presented approach is appropriate for investigating key research questions that only large-scale modelling efforts can help to answer, for example, those related to: global hotspot identification (Figure 4-6), the relative importance of different sectors across the globe (Figure 7) and meta-trends in water quality dynamics (e.g. Figure 10-11). For these analyses, we find it important to use globally consistent input data for DynQual runs made at the global scale (as per this paper), in order to facilitate meaningful comparisons across different world regions. Yet, this necessitates a simplified approach (see #8).

With these considerations, we choose to evaluate the global output from DynQual using metrics that focus on the residuals (i.e. prediction errors) using the normalized root mean squared error (nRMSE) (Figure 3, Figures S2a-5a); and by evaluating the ability of DynQual to simulate concentrations within a concentration range (Figure S6). This follows the evaluation approach adopted by global water quality models that are comparable to ours (e.g. Beusen et al., 2015; UNEP, 2016; van Vliet et al., 2021). Thanks for the good suggestion to add spatial patterns comparing the mean observed vs. modelled concentrations, these will supplement the existing evaluation approach nicely and so we will add these to the SI.

3) what is a good nRMSE value? It would be beneficiary to add the Nash–Sutcliffe model efficiency coefficient (NSE) which is a widely used dimensionless metric in hydrology and water quality literature.

Model evaluation spatial patterns presented in van Vliet et al. (2021) suggest acceptable performance below 100% (1.00) for total dissolved solids and biological oxygen demand. For large rivers (e.g. Rhine, Mississippi), Beusen et al., (2015) consider normalised RMSE (nRMSE) values of 50% (0.5) acceptable for nitrogen and phosphorus, in the view of the global scale of the model. Their model evaluation results including all rivers found a nRMSE of 124% (1.24) for N and 184% (1.84) for P, based on average annual concentrations, which they consider acceptable.

As also expressed in #4, the water quality parameter of interest is especially important to also consider here. For example, Reder et al., (2015) argue that as NSE and RMSE are sensitive to high extreme values (Moriasi et al., 2007), they need to be applied with caution in bacteria modelling where in-stream concentrations can vary across several order of magnitudes. The applicability of these metrics for model evaluation are also considered in the initial validation of DynQual in Jones et al., (2022), where it is shown for fecal coliform (FC) concentrations in some example stations (Figure 1) that while the model is generally capable of simulating concentrations within the correct concentration ranges, the magnitude of variability in the measured timeseries (also occurring over short time periods) is severely underestimated. High variability in observed FC concentrations is common across almost all monitoring stations, with 88% of stations reporting FC concentrations that range over three or more orders of magnitude (Jones et al., 2022).

We will further reflect on these points in the manuscript, supplemented with more comparisons to other global water quality models (as per #4).

[Figure]

**Figure 1.** Selected time series of observed vs. simulated fecal coliform (FC) concentrations (cfu/ 100mL). Black (daily) and red (rolling 30-day average) lines indicated simulated FC concentrations; whereas blue crosses are observed concentrations. As displayed in Jones et al., (2022).

4) this manuscript in general lack literature discussion or comparison in terms of model performance (e.g., Figure 3), for example, what is other water quality model performance in terms of nRMSE? There might be few global scale water quality models. But I guess it could be useful to add a few comparisons with other watershed-scale water quality models.

Our comparisons to other global water quality models regarding performance are currently done in the discussion section (lines 508 – 526). We agree that more comparisons to other global water quality models regarding performance can be added – both of terms of the comparisons made to the other global water quality models we already discuss (e.g. van Vliet et al., 2021; UNEP 2016; Wen et al., 2017), and also to other global water quality models (e.g. Beusen et al., 2015). This will be added to the revised manuscript. When comparing global water quality models (including model performance), inherent differences between model set-ups must also be acknowledged. With regards to this, areas that are especially important to consider related to the model are the: 1) spatial extent (e.g. lumped vs. distributed models); 2) temporal resolution (e.g. daily vs. monthly vs. annual vs. decadal); and 3) water quality constituent of interest (also see #3). We will also reflect on these points in detail in the discussion section.

However, we find difficulties in comparing our global water quality model output directly to watershed-scale water quality models, which are typically applied to investigate vastly different research questions and have different purposes. These models can also incorporate locally relevant input data which are lacking in global approaches, can be parameterized for local conditions and typically have observation data of good quality and record length for calibration and validation. However, the flexible model does allow for applying DynQual at different spatial scales, and with different configurations. For example, pollutant loadings can be calculated independently of the DynQual emissions module and applied as a forcing – which could be particularly beneficial where knowledge of pollutant emissions exceeds the globally-applicable datasets. (Calibrated) hydrological output can also be forced directly to DynQual, thus making use only of the routing module of PCR-GLOBWB2/DynQual. Future work will seek to investigate the potential of DynQual to answer research questions at finer spatial scales (e.g. watershed, country-level). Here, comparisons to existing watershed-scale models will be important and relevant.

5) Line 200, can the decay coefficient be specified by the user?

Yes, as with all parameters/ coefficients in the model setup, decay coefficients could be specified by the user directly in the source code. We will clarify this in the manuscript. In the manuscript, we present the way this has currently been implemented within DynQual (based off existing global water quality modelling work). These are displayed in Equations 4 (BOD) and 5 (FC), with parameter values for FC in Table 1.

6) Line 220, is it a constant background concentration or a time-varying background concentration through each timestep?

We use a constant background concentration for TDS. While time-varying background concentrations could technically be implemented into DynQual easily, the availability of data (especially at the global scale) to determine time-varying background concentrations is a limitation. For other time-varying sources of pollutant loadings to the surface water network, for example those related to highly seasonal irrigation regimes, these are implemented at the daily timestep. We will clarify this in the manuscript.

7) what was the computational time to run for 1-year simulation?

In short, this will largely depend on: 1) model configuration (Figure 1); and 2) the target spatial extent.

Global 5 arc-min DynQual runs that are coupled with PCR-GLOBWB2, as done in this work, have a wall-clock time of approximately 6 hours when run with parallelization, due to the requirement to run with the kinematic wave routing option. This is a more computational demand routing equation than e.g. travel-travel time characteristic routing option (Sutanudjaja et al., 2018), but provides greater realism which is needed for higher accuracy discharge and water temperature simulations. Our calculation times are more or less equivalent to the PCR-GLOBWB2 run times given in Sutanudjaja et al., (2018) using kinematic wave routing. So called "offline" global DynQual runs, which use hydrological input (i.e. baseflow, interflow and direct runoff) as an external forcing, are somewhat (~20%) quicker.

The parallelization strategy for global PCR-GLOBWB2/DynQual involves dividing the model domain into 53 groups of river basins that run independently of each other as 53 separate processes (see Sutanudjaja et al., 2018 for more details). The length of time it takes for each individual process to run is dependent on the size of the land mask (i.e. the number of pixels). Thus, for global parallelization runs, the computational time to run for a 1-year simulation is equal to that of the largest land mask. Yet, PCR-GLOBWB2/DynQual does not necessarily need to be run at the global extent – users could alternatively select a particular river basin(s) or define their own land mask. The self-contained example for the Rhine basin that we provide (https://zenodo.org/record/7027242#.Y8fHOhfMJPY) takes approximately 45 minutes to run for 1 year.

8) Supplement Line 295, does it mean reaction is underestimated compared to discharge (dilution)?

The current model set-up for DynQual emissions module is to quantify loadings at the individual gridcells where the pollution activity is located (e.g. populations), with loadings entering the surface water network at the same location (which are subsequently routed through the surface water stream network based on the flow direction (routing) map; Figure 2). This approach is advantageous in that we can use globally available datasets for estimating pollutant emissions (e.g. populations). The ability to use globally consistent input data is important for DynQual runs made at the global scale (in line with the focal point of this paper), whereby we want to facilitate meaningful comparisons across different world regions.

This approach has the disadvantage of spatial mismatches between the generation of pollutant loadings and the actual locations where loadings enter the stream network occur. Simulated concentrations in gridcells with low water availability – i.e. headwater streams - are particularly sensitive to this, and concentrations in these gridcells are typically masked in global water quality studies. Similarly, point sources of pollution (e.g. wastewater treatment plants) typically discharge into higher-order streams, the location of which might not exactly coincide with population density. DynQual tracks the mass of pollutants (not concentrations) in discrete gridcells per timestep (Figure 1). Thus, these masses are transported downstream to higher-order streams whereby the pollutant mass will be more feasible with respect to the dilution capacity – hence the better model performance in gridcells with greater water availability.

9) Supplement Line 300, what is high data availability, and how many data points during 1980-2019?

"High data availability" is, of course, relative. In this paper, and as per Jones et al., (2022), high data availability refers to >30 measurements in the target time period. We will add this detail to the SI. In general, water quality observations are limited in both geographical space and fragmented across time . There are also issues related to data access, with data collected by both governmental organizations and private institutions not being made publicly accessible. These are key motivations for conducting large-scale water quality modelling efforts, such as DynQual, i.e. to provide insights on water quality status in data scare regions such as Africa and large parts of Asia.

10) Supplement Line 305, Figure S3 (b, c) what are the nRMSE and NSE values for these two rivers? It seems that the model overestimated a lot for peaks.

For Figure S3b, the nRMSE is 0.15, with a mean observed concentration of TDS 399 mg l$^{-1}$ (349 observations) and a mean modelled concentration of TDS 388 mg l$^{-1}$. For Figure S3c, the nRMSE is 0.24, with a mean observed concentration of TDS 25 mg l$^{-1}$ (468 observations) and a mean modelled concentration of TDS 24 mg l$^{-1}$. We wanted to explicitly include some time-series plots in our evaluation to demonstrate the dynamic nature of the model simulations, and selected stations based on the high data availability and the long-time series of observations.

However, comparing individual (instantaneous) observed concentrations vs. simulated concentrations comes with challenges at large scales – particularly when it comes to min and max concentrations. As demonstrated in ~2007 in the Drammenselva river, observed TDS concentrations fluctuate by a factor of 2 over short time periods (~25 mg l$^{-1}$ to 50 mg l$^{-1}$), with the peak being entirely missed in our simulations. This is difficult to attribute (and is beyond the scope of DynQual), given the global scale of our model and the underlying assumptions – perhaps it is a missing emission process (e.g. delivery of road-salts to the stream network) or a temporal mismatch in the simulated vs. actual hydrological conditions (i.e. dilution effect). Yet, max TDS concentrations in the observed and simulated time-series for the Drammenselva river are very similar (~50 mg l$^{-1}$). Max simulated concentrations (~600 mg l$^{-1}$) are indeed somewhat higher than observed concentrations (~500 mg l$^{-1}$) in the Kalamazoo river. Again – difficult to attribute – this could similarly due to a bias in the dilution component of our model or due to an overestimation of TDS loadings originating from short-lived processes (e.g. those from urban surface runoff). As described in #2, the overarching purpose of the model must be considered, and please note that we are not using DynQual to draw conclusions regarding extremes (min-max concentrations).

For reasons described in #2&3, we refrain from calculating NSE values, which have not been used for evaluating the performance of any global water quality model to date.

**References**

Beusen, A. H. W., Van Beek, L. P. H., Bouwman, A. F., Mogollón, J. M., & Middelburg, J. J. (2015). Coupling global models for hydrology and nutrient loading to simulate nitrogen and phosphorus retention in surface water–description of IMAGE–GNM and analysis of performance. Geoscientific model development, 8(12), 4045-4067.

Hrachowitz, M., Savenije, H. H. G., Blöschl, G., McDonnell, J. J., Sivapalan, M., Pomeroy, J. W., ... & Cudennec, C. (2013). A decade of Predictions in Ungauged Basins (PUB)—a review. Hydrological sciences journal, 58(6), 1198-1255.

Moriasi, D. N., Arnold, J. G., Van Liew, M. W., Bingner, R. L., Harmel, R. D., & Veith, T. L. (2007). Model evaluation guidelines for systematic quantification of accuracy in watershed simulations. Transactions of the ASABE, 50(3), 885-900.

Reder, K., Flörke, M., & Alcamo, J. (2015). Modeling historical fecal coliform loadings to large European rivers and resulting in-stream concentrations. Environmental Modelling & Software, 63, 251-263.

Sutanudjaja, E. H., Van Beek, R., Wanders, N., Wada, Y., Bosmans, J. H., Drost, N., ... & Bierkens, M. F. (2018). PCR-GLOBWB 2: a 5 arcmin global hydrological and water resources model. Geoscientific Model Development, 11(6), 2429-2453.

UNEP: A Snapshot of the World's Water Quality: Towards a global assessment, United Nations Environment Programme, Nairobi, Kenya, 162pp, 2016.

van Vliet, M. T., Jones, E. R., Flörke, M., Franssen, W. H., Hanasaki, N., Wada, Y., & Yearsley, J. R. (2021). Global water scarcity including surface water quality and expansions of clean water technologies. Environmental Research Letters, 16(2), 024020.

Voß, A., Alcamo, J., Bärlund, I., Voß, F., Kynast, E., Williams, R., & Malve, O. Continental scale modelling of in-stream river water quality: a report on methodology, test runs, and scenario application. Hydrological Processes, 26(16), 2370-2384.

Wanders, N., van Vliet, M. T., Wada, Y., Bierkens, M. F., & van Beek, L. P. (2019). High-resolution global water temperature modeling. Water Resources Research, 55(4), 2760-2778.

Wen, Y., Schoups, G., & Van De Giesen, N. (2017). Organic pollution of rivers: Combined threats of urbanization, livestock farming and global climate change. Scientific reports, 7(1), 1-9.

---

## Author Comment (AC2)

**Response to Reviewer 1 (gmd-2022-222)**

The manuscript provides a new global model for water quality assessment (DynQual). The model runs at a daily time step for the grid cell of approximately 10km by 10 km at the equator. The authors made a major effort to develop such a comprehensive dynamic model on a global scale. One of the model's strengths (compared to other existing large-scale water quality models) is the sector contributions (e.g., livestock, irrigation, manufacturing) to surface water pollution per grid and day while considering dynamics in pollutant routing through the river network. The model is process-based and largely uncalibrated. It makes the model more flexible to represent the processes based on characteristics (e.g., livestock number, people, runoff) and less dependent on observations (which are often scarce). In this way, there is an opportunity to apply the model for the future while considering the future characteristics of the areas. Validating the global model is not easy. The authors did a great job here and managed to validate in a good way. The manuscript is well-written, with beautiful maps and graphs. It is easy to follow the description of the model and results.

I suggest a minor revision. Below, I provide suggestions that can be helpful to improve the manuscript:

We thank the reviewer for these suggestions. The overarching theme of the comments is to further increase clarity in some sections (particularly methods and discussion). As noted by the reviewer, some of these additional descriptions are currently contained in the SI. We fully agree that some improvements can be made/ pertinent information can be moved from the SI on these points in order to further strengthen the manuscript.

Please find our point-by-point response to the detailed reviewer comments below. The more substantial additions or changes we will make to the text are indicated in this documents in *"italics"*. For the more minor and editorial changes, we indicate the actions we will take in purple. Full details of these changes will be available in a manuscript with tracked changes that we will submit once the discussion period has ended.

1. Abstract: It is a nice abstract, but does not provide any insights that we learn from the model application. Please add the main messages (2-3 sentences) that reflect the two main objectives of the model application:
   1. Pattern and trends
   2. Sector contributions

True. We will add the following text to the abstract:

*"Modelled output indicates that multi-pollutant hotspots are especially prevalent across northern India and eastern China, but that surface water quality issues are present across all world regions. Trends towards water quality deterioration are most profound in the developing world, particularly Sub-Saharan Africa."*

Given this manuscript is submitted as a model description paper, and due to the broad scope of this manuscript: 1) introducing the DynQual model; and 2) showcasing a variety of potential applications (e.g. spatial patterns, trends, seasonality and sector-specific contributions) for multiple pollutants and at the global scale, we refrain from including more specific (e.g. pollutant specific) results from the model application – preferring the more general outlook as summarised in the text above.

2. Methods: They are described rather concisely. Details are provided in the supplementary materials, which is nice. Nevertheless, I have four suggestions to elaborate on:
   1. Figure 1: please add the legend. The description of what colors and different arrows (dashed and solid) mean is not clear;

Thanks for this. We will add a legend to Figure 1.

   2. Pollutant loadings to streams: this is well described in the supplementary information, but very concisely described in the main methods. Please add a few more sentences to tell the reader how pollutant loadings are calculated (e.g., the summary of the description from the SI). You can elaborate on the text where you mention the mass-balance approach. Here, you can briefly tell that pollutant loadings from livestock activities are simulated as a function of livestock number, excretion rates of pollutants per animal and day, and removal of pollutants during waste management practices while considering runoff from land to streams (see SI Section 1.4 for details). A similar description can be given for other sectors.
   3. Sector contribution: you do mention sectors in the methods, but briefly. Please elaborate more on what sectors exactly include and how. For example, the irrigation sector: what does it include? Which crops? rainfed irrigation? Livestock sector: which animals? I do know that some details are in the SI. But I do feel a need to give a bit more description of the main methods of the manuscript.

As noted by the reviewer, the additional descriptions on pollutant loading estimates (which directly also relate to the sectors) are currently contained in the SI. It was a deliberate choice to keep this section relatively short in the manuscript – due to the fact that pollutant loadings can alternatively be forced directly to DynQual. This provides an option for users who have pre-defined (or prefer to use their own methodology for estimating) pollutant loadings, yet still wants to make use of the pollutant routing and in-stream decay components of DynQual (see point 1.2). Nevertheless, we agree with the reviewer that some pertinent information on the estimation of pollutant loadings should indeed be included in the manuscript.

We will substantially expand section 2.3 to summarise the key information with respect to pollutant loading estimates for each sector, with the following text:

*"Loadings from the domestic sector are estimated by multiplying gridded population numbers with region-specific per capita excretion rates. For the manufacturing sector, a mean effluent concentration is multiplied by gridded estimates of return flows from the manufacturing sector. Gridcell specific urban surface return flows are simulated directly by PCR-GLOBWB2 and are multiplied by a region specific mean urban surface runoff effluent concentration. Gridded livestock numbers for buffalo, chickens, cows, ducks, goats, horses, pigs and sheep are multiplied by pollutant excretion rates per livestock type and by region. The livestock sector is sub-divided into 'intensive' and 'extensive' production systems based on livestock densities, in order to better account for differences in the paths by which waste enters the stream network. TDS loadings from the irrigation sector are estimated by multiplying irrigation return flows simulated by PCR-GLOBWB2 with spatially-explicit mean irrigation drainage concentrations based on electrical conductivity over the top- and sub-soil. Return flows from thermo-electric powerplants are included as a source of heat pollution by considering the temperature difference between these flows and ambient conditions."*

Through also including a new table in the manuscript as per your suggestion (Table 2, see below), we hope also make the required input data for estimating pollutant loadings within a DynQual run more clear.

> 4. Downscaling: for example, some of the input data (e.g., livestock numbers) is regional, but DynQual requires the grid cell data. How did the authors go from regional to the grid cell, but also from annual to daily levels? Which model inputs require scaling (e.g., annual->daily; regional -> grid cell)? and which did not. This is not well elaborated. I suggest adding a few sentences on this in the main methods and giving more details in the SI. I suggest adding an overview table showing the list of model inputs and indicating which ones were aggregated from region to grid and from annual to daily.

We agree that an overview table that describes: 1) the list of model inputs per water use sector; 2) data source and 3) spatio-temporal resolution is a valuable addition to the manuscript, with the elaborated details per sector in the SI (sections 1.1 – 1.6).

We will include this in the manuscript:

**Table 2.** Summary of key input data used for the estimation of pollutant loadings in the presented application

| Sector | Data | Source | Spatio-temporal resolution |
|---|---|---|---|
| Domestic | Population | (Lange and Geiger, 2020) | 5 arc-min; annual |
| | Excretion rates | (UNEP, 2016; Van Vliet et al., 2021) | Regional; constant |
| Manufacturing | Manufacturing return flows | PCR-GLOBWB2 (*simulated*) | 5 arc-min; daily |
| | Effluent concentrations | (UNEP, 2016; Van Vliet et al., 2021) | Global; constant |
| Urban surface runoff | Urban surface runoff | PCR-GLOBWB2 (*simulated*) | 5 arc-min; daily |
| | Effluent concentrations | (UNEP, 2016) | Regional; constant |
| Livestock | Livestock populations | (Gilbert et al., 2018) | 5 arc-min; annual |
| | Excretion rates | (Weaver et al., 2005; Wilcock, 2006; Robinson et al., 2011; Wen et al., 2017; Vigiak et al., 2019; Van Vliet et al., 2021) | Regional; constant |
| Irrigation | Irrigation return flows | PCR-GLOBWB2 (*simulated*) | 5 arc-min; daily |
| | Effluent concentrations | (Batjes, 2005) | 30 arc-min; constant |
| Power | Power return flows | (Lohrmann et al., 2019) | 5 arc-min; annual |
| | $\Delta T$ | (Van Vliet et al., 2012a) | Global; constant |

We will add some small clarifications to these sections in line with the reviewers suggestions – such as for the downscaling procedure (where appropriate).

With respect to livestock numbers explicitly, these were sourced at 5 arc-min resolution directly from Gilbert et al., (2018) for a reference year of 2010. Thus, no additional downscaling was required on this data. What was required was to account for temporal changes to livestock numbers over past years, for which we had to make a coarse assumption (regional-scale percentage changes applied equally across all gridcells within that region) based on FAO data on changes to livestock numbers in the past decades (Thomson, 2003). We will more explicitly state this in the SI and acknowledge this as an uncertainty in both the manuscript and SI.

3. Discussion: It is very concise and to the point. Some aspects can be expanded and a few aspects can be added:

1. Comparison with other studies: the authors do this for the pollutants that they consider. I also think that the manuscript will benefit if the authors add comparisons in terms of modeling approaches, pollution hotspots, and sector contributions (what new aspects are added in this DynQual model and what new aspects we learn from the model application compared to other models). The authors may consider expanding the discussion (a few sentences) on comparing their pollution hotspots not only for TDS, BOD, and FC but also for other pollutants as well because pollution hotpots often match between pollutants.

Thanks for this suggestion. We agree that a short section comparing the results from our model application (e.g. pollution hotspots) to other studies is a valuable addition to the manuscript, and will add the following text:

*"Overall, our modelled spatial patterns in surface water quality match well with previous regional and global assessments – displaying multi-pollutant hotspots (e.g. TDS, BOD, FC) to be located across northern India and eastern China in particular (UNEP, 2016; Wen et al., 2017; Van Vliet et al., 2021). As demonstrated here, DynQual can be used to further enhance the understanding of the main drivers of pollutant via (dynamic) sectoral attribution. The dynamic nature of DynQual also facilitates analysis of intra- and inter- annual trends in surface water quality. Consistent with a recent data driven approach (Desbureaux et al., 2022), albeit for different water quality constituents (e.g. total phosphorus) we find a general trend towards surface water quality improvement in development countries and deterioration in developing countries."*

2. Implications of the limitations: any models have limitations. DynQual has as well. Examples are livestock numbers in extensive and intensive production systems that do not vary among days, excretions rates of pollutants in manure, and human waste that are constant across the days and within the regions. I understand that sources such as open defecation, and direct discharges of manure to rivers are not considered. It is fine, but this needs to be discussed. It is important to give examples of the main limitations and reflect critically on their implications on the main conclusions of the manuscript.

Thanks for this comment. We will expand our section on model uncertainties to include more examples, including the example suggested by the reviewer with respect to constant livestock numbers.

*"Being a global model, DynQual is inherently unable to represent all aspects relevant to the local context. For example, the lack of information on TDS emissions from mining activities and road de-icing. Livestock numbers in both intensive and extensive production systems are constant across the days of the year, with changes to livestock numbers only applied annually and based on regional averages, without consideration of pollutant retention in soils. Spatial mismatches between the generation of pollutant loadings and the location of discharge to the stream network can result in the simulation of unrealistic concentrations, particularly in gridcells with very low water availability (i.e. headwater streams). This can occur, for example, where the drivers of point-source pollutant emissions (e.g. population) do not directly coincide with the location of wastewater treatment plant outlets."*

We will also include a section to more critically evaluate the overarching purpose and applications of these type of global scale models, providing examples of the main type of research questions that DynQual can help to investigate (see #3.3).

3. The usefulness of the model: DynQual has many useful applications (e.g., trends, patterns, future analyses, etc). The authors briefly mention this in paragraph 545. I think the authors can better emphasize how useful their model is compared to other models. For example: which scientific questions we can answer with this model that we could not answer with the previous models? The authors could add a few sentences on this in paragraph 545.

Agreed. We will include some more text in the discussion to further emphasise the types of research questions DynQual can help to address, with specific reference to the results (Figures) we present in the manuscript:

*"The presented application of DynQual allows for the investigation of research questions that only large-scale modelling efforts can address, including: global hotspot identification (Figures 4 - 6), the relative importance of different sectors across the globe (Figure 7) and meta-trends in water quality dynamics (Figures 10 – 11). Our approach has particular value for simulating surface water quality in ungauged catchments, and our use of a globally consistent input data facilitates meaningful comparisons of surface water quality across different world regions."*

4. Supplementary information: It is well written. I have three suggestions:

  1. Units: they are missed in some equations. For example, units are not included for the following variables: $R_{dom,i,n}$ and $Pop_n$ (population per km$^2$? Total population?) in equation [1], $L_{man,i,n}$ and $R_{man,i,n}$ in equation [2], $L_{urb,i,n}$ and $R_{USR,i,n}$ in equation [3]. Please also check the variables in equations [4] and [5] and add units for every variable. This will avoid misinterpretation.

Thanks for highlighting this. We will thoroughly go through both the manuscript and the SI to check this. We will also make some small corrections to remove inconsistencies in terms used across the two documents (e.g. in the manuscript equations we denoted pollutant using *i* but in the SI using *p* – we now consistently use *p* across both documents).

We will also add the equations for estimating removal of pollutants at wastewater treatment facilities per sector (e.g. $R_{dom,i,n}$, $R_{man,i,n}$). As these have been described extensively in previous work, we refrain from providing a full detailed description of these again here - yet we agree it is important to present the equations with the terms again here. We will also amend equations 3 and 4 as necessary to avoid misinterpretation.

  2. Livestock activities: is the number of livestock the same per day? Is this number per km$^2$ or ha? Did you consider soil processes and associated retentions of the pollutant in soil when you calculate loadings into the streams? All this was not very clear to me. Please clarify the description of equation 4.

Gilbert et al., (2010) provide a snapshot of global livestock numbers (for the 8 livestock types considered) with a reference year of 2010. We then apply estimates of annual change (at regional spatial scale) to livestock numbers in the past and future based on statistics from the FAO to estimate livestock numbers for the other timesteps. We will alter our sentence in the SI to make this more explicit.

Thus, yes, the number of livestock remains constant per day throughout the year. Soil processes and associated retentions of pollutants in soil are not explicitly accounted for, only

the transport of pollution from this sector to streams via surface runoff. These are limitations which we will also now explicitly state in the discussion section of the manuscript (as per review comment #3.2), and areas for improvement in estimates of pollutant loadings from the livestock sector within DynQual.

3. Scaling: please elaborate on which input data required to be scaled from region to grid and from annual to daily, and how this was done.

Due to the significant overlap, we have addressed this comment in our response to comment #2.4 (i.e. Table 2). Please see above.

---

## Author Comment (AC3)

**Response to Reviewer 2 (gmd-2022-222)**

In addition to (or as an emphasis of) the community comment by Jason Ke and the comments from Referee #1 I have the following suggestions for improving the paper.

Thanks for these comments. As per the other two reviewers, please find our point-by-point response below. We indicate areas where we intend to take action in purple. For the more substantial additions or changes we intend to make to the text, we have copied a draft of these changes into this document in *"italics".* Full details of these changes will be available in a manuscript with tracked changes that we will submit once the discussion period has ended.

- Model description paper in GMD should focus on a detailed description of the scientific basis of a model and the technical/ numerical implementation. In the current version of the manuscript, there is a certain imbalance between the respective contents in the main paper and the supplemental material. Most of the description of model details in the supplemental material should be directly mentioned in the main paper.

We agree with this and your subsequent comment that the balance of the manuscript can be improved, and that some more technical details should be added to the manuscript. We have summarised the information we intend to add to the manuscript below under three aspects: 1) surface water and pollutant routing; 2) pollutant loading calculations; and 3) other technical details regarding model code, data format and running time:

1. Surface water and pollutant routing

We previously referred only to the PCR-GLOBWB2 documentation (*Sutanudjaja et al., 2018*) for these descriptions to avoid too much overlap. However, we agree that specific information regarding the routing routine should also be included here, and as such we will add the following text to the manuscript:

*"The routine for surface water and pollution routing follows an eight-point steepest-gradient algorithm across the terrain surface (local drainage direction) in a convergent drainage network with the lowermost cell connected to either the ocean or an endorheic basic (Sutanudjaja et al., 2018). Routing within DynQual uses the kinematic wave approximation of the Saint-Venant equations with flow described by Manning's equation, solved using a time-explicit variable sub-time stepping scheme based on the minimum Courant number (Sutanudjaja et al., 2018). In the coupled configuration, surface waters are subject to water withdrawals and return flows from the domestic, industrial, livestock and irrigation sectors calculated within the water use module of PCR-GLOBWB2."*

2. Pollutant loadings

As per our response to Reviewer #1, it was a deliberate choice to keep the section on pollutant loading calculations relatively short in the manuscript given that pollutant loadings can alternatively be forced directly to DynQual. Nevertheless, we agreed with the reviewer that some pertinent information on the estimation of pollutant loadings should indeed be included in the manuscript. We will expand section 2.3 to summarise the key information with respect to pollutant loading estimates for each sector:

*"Loadings from the domestic sector are estimated by multiplying gridded population numbers with region-specific per capita excretion rates (SI Section 1.1; Table S1). For the manufacturing sector, a mean effluent concentration is multiplied by location specific gridded*

*estimates of return flows from the manufacturing sector (SI Section 1.2; Table S2). Urban surface return flows are approximated by multiplying surface runoff (simulated by PCR-GLOBWB2) with the gridded urban fraction, which are multiplied by a region-specific mean urban surface runoff effluent concentration (SI Section 1.3; Table S3). The livestock sector is sub-divided into 'intensive' and 'extensive' production systems based on livestock densities to better account for differences in the paths by which waste enters the stream network (SI Section 1.4, Table S4). Gridded livestock numbers for buffalo, chickens, cows, ducks, goats, horses, pigs and sheep are multiplied by pollutant excretion rates per livestock type and by region (SI Section 4, Table S5 – S7). The livestock sector is sub-divided into 'intensive' and 'extensive' production systems based on livestock densities to better account for differences in the paths by which waste enters the stream network (SI Section 1.4; Table S4-S7). TDS loadings from the irrigation sector are estimated by multiplying irrigation return flows simulated by PCR-GLOBWB2 with spatially-explicit mean irrigation drainage concentrations based on salinity (as indicated by electrical conductivity) over the top- and sub-soil (SI Section 1.5). Thermal effluents (heat dumps) from thermoelectric powerplants are included as a point sources of advected heat by considering the temperature difference between the return flows and ambient surface water temperature conditions (SI Section 1.6). Pollutant loadings from the domestic, manufacturing and intensive livestock sectors, and from urban surface runoff, can be abated based on gridcell-specific wastewater practices. The proportion of pollutant loadings removed by wastewater treatment practices is estimated by multiplying the fraction of each treatment level occurring in a gridcell by the pollutant removal efficiency associated with that treatment level, as described in detail in previous work (Jones et al., 2021; Jones et al., 2022)."*

As also per the request of Reviewer #1, we will also add a Table summarising the required input data for pollutant loading emissions within a DynQual run.

3.  Other technical details

We will add some sentences to the manuscript to improve the description of the technical details:

*"As per PCR-GLOBWB2 (Sutanudjaja et al., 2018) and DynWat (Wanders et al., 2019), DynQual is written in Python and is run using an initialization (.ini) file in which key aspects of the model run are defined (e.g. spatial extent, simulation period, paths to parameter and forcing files). Most input files required and all output are in NetCDF format. Global 5 arc-min DynQual runs that are coupled with PCR-GLOBWB2 have a wall-clock time of approximately 6 hours per year when run with parallelisation, due to the requirement to use the kinematic wave routing option for higher accuracy discharge and water temperature simulations. This is equivalent to the PCR-GLOBWB2 run times given by Sutanudjaja et al., (2018). DynQual runs performed in the stand-alone configuration are faster (~20%)."*

-   Another large part of model description papers should be dedicated to the model verification. This aspect is quite underrepresented in the current version of the manuscript. Model verification is only presented in line 304 to 321 and one figure in the main paper. Some additional results are included in the supplemental material but without a thorough presentation, analysis and discussion of the results. These aspects need to gain much more space in the main paper as this is a central part of model description papers in GMD.

We agree with you and Community Comment #1 that model evaluation is under-represented in the current submission. We also agree with the comment from Reviewer #1 that

"validating the global model is not easy" – comparing individual (instantaneous) observed concentrations vs. simulated daily concentrations does indeed come with challenges at large scales. Data availability, both in terms of spatial coverage and the number of observations per water quality monitoring station, also presents significant challenges. Lack of global spatial coverage impacts validation efforts for all four water quality constituents, while data availability issues is particularly limiting for BOD and FC. This is somewhat of a "catch-22" for large-scale water quality modelling efforts – poor data availability across space and time is a key motivation for developing physically-based water quality models (to fill in these data gaps), but also is a severe limitation for both model development and evaluation.

We appreciate the praise from Reviewer #1 regarding our current efforts here, yet we also agree with Community Comment #1 and Reviewer #2 that this section can (and should) be further improved and expanded. To give the validation aspects more emphasis in the revised manuscript, we intend to assign "Model evaluation" into its own sub-section and expand on this section both with additional analysis (including a new figure) and discussion.

- Along with a more detailed model verification, the authors also need to emphasize the discussion on possible model limitations.

Agreed. As raised by Reviewer #1, we will add an additional paragraph with examples of specific uncertainties and model limitations of DynQual:

*"Uncertainties in surface water quality simulations arise from a combination of uncertainties associated with quantifications of pollutant loadings (e.g. pollutant excretion, emission rates and sector-specific return flows), the quality of hydrological simulations (e.g. discharge and velocities) and the representation of in-stream processes (e.g. decay coefficients). These uncertainties are amplified when modelling at large spatial extents. In-stream pollutant concentrations are highly sensitive to dilution capacity, thus the quality of the river discharge simulations. This issue contributes to uncertainties in simulated concentrations particularly in headwater streams. Fixed estimates of decay coefficients have been assumed, which contributes to uncertainties in simulations of reactive constituents such as BOD and FC. In addition, the representation of lakes and reservoirs in DynQual is rudimentary, with total (routed) loadings instantaneously averaged over the volume of the water body assuming full mixing."*

*With respect to pollutant loading quantifications, spatial mismatches between the generation of pollutant loadings and the location of entry to the stream network (return flows) can result in the simulation of unrealistic concentrations, particularly in gridcells with very low water availability (i.e. headwater streams). This can occur where the drivers of point-source pollutant emissions (e.g. population) do not directly coincide with the location of wastewater treatment plant outlets. A lack of temporally-explicit input data can hinder proper representation of sectors with strong inter-annual variability - a notable limitation for the livestock sector. Here, simplified assumptions are required for aspects such as livestock population numbers (assumed to be constant across days of the year), change to livestock numbers across multi-year periods (applied annually and based on regional averages) and transportation pathways to the stream network (assumed to be a function of surface runoff excluding the representation of processes that affect pollutant retention in soils). Locally relevant sources of pollution may also be entirely excluded, such as the lack of information on TDS emissions from mining activities and road-deicing.*

We also intend to add a paragraph to the discussion which will highlight the key limitations and challenges of large-scale water quality modelling more generally, and further emphasise the types of research questions these approaches can help to address.

- A much larger portion of the manuscript (lines 323-491) is dedicated to the presentation of spatial patterns and trends of (long-year global) simulation results. Section 3 is named "model demonstration" and the model evaluation part (see above) is only a small part of this section. Showing possible applications of the model is, of course, interesting but model fidelity has do be demonstrated first and in much greater detail before spatial patterns and trends can be presented. Hence, there needs to be a better balance between the "model evaluation" and the "model demonstration" part.

In line with one of the previous comments, we propose to split the sub-section "Model run set-up and validation" into two separate sections: i.e. 3.1) Model run setup and 3.2) Model evaluation – and significantly expand on these sections in line with the recommendations.

We agree that, as a model description paper, the manuscript needs to be more balanced. We propose to remove results related to the average annual fluctuations (Figure 8 + associated description) and the sector-specific time series (Figure 11 + associated description) from the manuscript, instead keeping the focus on spatial patterns and trends.

Combined, we believe these efforts will bring better balance to the manuscript in line with requirements for GMD model description papers.

- The performance of DynQual should also be discussed in comparison to other water quality models, e.g. available catchment or regional scale models. The level of detail in terms of available input data and spatio-temporal resolution is, of course, different but the modelling community and potential users need to know what quality of output they could expect from this newly developed model compared to existing ones. Although the scope of the model might be a bit different, the interpretation of results needs to rely on the closeness to observation data (i.e. the prediction capability) and this one needs to be compared to already existing modelling approaches.

We agree that the performance of DynQual should be discussed in comparison to other large-scale water quality models. We included some comparisons already to the most comparable studies (e.g. to van Vliet et al, 2021, Wen et al., 2017 and UNEP 2016). However, we will (re-)review the published large-scale water quality literature to try and expand this discussion further.

We find difficulties into statistically comparing the performance of an uncalibrated global water quality model (with global parameterisation and input data sets) to watershed specific water quality models. These two types of models have fundamentally different purposes. Watershed-scale models can incorporate locally relevant input data and processes which are impossible to meaningfully represent in global approaches given data limitations. Watershed models are parameterized for specific local conditions and typically are calibrated based upon observation data of good quality and record. This issue is also addressed in our response to Community Comment #1 Q4. We will more specifically allude to the key differences between watershed vs. global surface water quality models in the manuscript:

*"Comparatively, watershed-scale surface water quality models can better incorporate locally relevant input data and processes, can be parameterized specifically for local conditions and*

*typically have observation data of good quality and record length for calibration and validation. This allows for higher precision and accuracy in both hydrological and water quality simulations, particularly with regards to the magnitude and timing of high and low flows and concentrations, a primary aim of watershed-scale models. However, these watershed-scale models are reliant upon detailed local knowledge which is severely lacking for many (particularly ungauged) catchments worldwide (e.g. large parts of Africa). Despite their limitations, process-based large-scale water quality models can facilitate first-order assessments of global water quality dynamics that are consistent across both space and time, such as those demonstrated in the model application section of this study."*

As also requested by Reviewer #1, we will also more specifically detail the usefulness of DynQual with respect to the types of scientific questions we can use it to address:

*"The presented application of DynQual allows for the investigation of research questions that only large-scale modelling efforts can address, including: global hot- and bright- spot identification (Figures 4 - 6), the relative importance of different contributing sectors to water quality status across the globe (Figure 7) and meta-trends in surface water quality dynamics (Figures 8 – 10)."*

- It is unclear, why the normalized RMSE was chosen as the sole performance measure for model evaluation. Other performance measures such as the Nash-Sutcliffe efficiency (NSE) or the Kling-Gupta efficiency (KGE) are much more common in model benchmarking and allow a clearer interpretation of the model performance, e.g. NSE lower or greater than zero. In addition, KGE even combines different aspects of model performance.

Our validation approach follows the standard practice that has been adopted in evaluating comparable large-scale water quality models in terms of both 1) modelling approach; and 2) purposes (e.g. Beusen et al., 2015; UNEP, 2016; van Vliet et al., 2021). It was for this reason that we chose to focus the evaluation of DynQual using metrics such as the normalized root mean squared error (nRMSE) and by evaluating the ability to simulate concentrations within a concentration range. As discussed above, we will revisit the water quality literature to improve our model evaluation section.

Please also note that no existing large-scale water quality model has used the NSE or KGE for model evaluation, aside from for water temperature (e.g. van Vliet et al., 2011). Please also see our response to Community Comment #1 Q3&10 with regards to the evaluation of large-scale water quality model output using NSE.

- It is mentioned that DynQual can be run in two modes: (1) coupled to PCR-GLOBWB2 and (2) in offline-mode with any other hydrological model. Please describe in more detail the technical aspects of the coupling between DynQual and PCR-GLOBWB2 as well as the (technical) requirements for using DynQual with other hydrological models, e.g. what are the required input data from the hydrological model and in which form they need to be provided to DynQual (e.g. netCDF files and their format).

We will add a sentence specifying the required input data to run the stand-alone configuration (please note this information is also contained in Figure 1):

*"…either: 1) in a stand-alone configuration with specific discharge (i.e. baseflow, interflow and direct runoff in m day$^{-1}$) fed from any land surface or hydrological model."*

To clarify input data formats, and also output formats, we will add a short statement to the manuscript:

*"Most input files required and all output are in NetCDF format."*

To elaborate on the coupling we will include a short sentence to describe this:

*"In the coupled configuration, surface waters are subject to water withdrawals and return flows from the domestic, industrial, livestock and irrigation sectors calculated within the water use module of PCR-GLOBWB2."*

- Time-stepping: Some lower bounds for the sub-daily time steps are mentioned (line 184ff). Does the model ensure that numerical stability criteria (Courant number, Peclet number) are met for the reactive transport equations?

Additional information will be added to the manuscript regarding the time-steps (and, more generally, the routine for surface water and pollutant routing):

*"The surface water routing routine follows an eight-point steepest-gradient algorithm across the terrain surface (local drainage direction) in a convergent drainage network with the lowermost cell connected to either the ocean or an endorheic basic (Sutanudjaja et al., 2018). Routing within DynQual uses the kinematic wave approximation of the Saint-Venant equations with flow described by Manning's equation, solved using a time-explicit variable sub-time stepping scheme based on the minimum Courant number."*

- Equation 9 for nRMSE: Please double-check this equation, the square is missing

We will correct this equation in the SI.

---

## Author Response (AR1)

**Response to reviewers: DynQual v1.0: A high-resolution global surface water quality model (gmd-2022-222)**

We thank the reviewers and community for taking the time and effort to critically evaluate our manuscript and SI, and for their valuable comments. We are pleased that the submission has overall been assessed to be novel, interesting and easy to follow – also with positive comments regarding our data analysis and presentation.

The key concerns raised (primarily by Reviewer #2 and Community Comment #1) are related to model evaluation, in addition to the overall balance of the manuscript in presenting model evaluation versus model demonstration results. We agree that more emphasis should be placed on model evaluation and made substantial revisions to : 1) include more evaluation (i.e. model validation) results; and 2) better describe and discuss the model validation results in more detail in the main text of the manuscript. We also feel it necessary to better discuss the challenges and limitations of validating global water quality models, as highlighted by Reviewer #1. In addition to these actions, we have better balanced the manuscript by being more selective in the figures and text we use in our model application sections.

The need to better elucidate the scope, aims and rationale of DynQual in the manuscript is also apparent to us, especially for readers not originating from the large-scale water quality modelling community. To address this (and in overlap with some other reviewer comments), we felt it necessary to add better descriptions of: 1) the strengths of this modelling approach; 2) the limitations of this approach; and 3) the types of research questions (and not) that these kind of high-resolution large-scale water quality models are suitable for addressing.

Please find our point-by-point response to the two reviewer comments, in addition to the community comment, below. Please note there is significant overlap in some of the comments, and hence our responses. For completeness we have responded to all comments independently, however where there is significant overlap, we refer to the review comment number that addresses that topic. We indicate the substantial additions and changes to the text in this document, together with line numbers (tracked changes version), in *"italics"*. Full details of all changes made can be viewed in the updated version of the manuscript that includes tracked changes.

**Response to Reviewer #1**

The manuscript provides a new global model for water quality assessment (DynQual). The model runs at a daily time step for the grid cell of approximately 10km by 10 km at the equator. The authors made a major effort to develop such a comprehensive dynamic model on a global scale. One of the model's strengths (compared to other existing large-scale water quality models) is the sector contributions (e.g., livestock, irrigation, manufacturing) to surface water pollution per grid and day while considering dynamics in pollutant routing through the river network. The model is process-based and largely uncalibrated. It makes the model more flexible to represent the processes based on characteristics (e.g., livestock number, people, runoff) and less dependent on observations (which are often scarce). In this way, there is an opportunity to apply the model for the future while considering the future characteristics of the areas. Validating the global model is not easy. The authors did a great job here and managed to validate in a good way. The manuscript is well-written, with beautiful maps and graphs. It is easy to follow the description of the model and results.

I suggest a minor revision. Below, I provide suggestions that can be helpful to improve the manuscript:

We again thank the reviewer for their comments and the helpful suggestions. The overarching theme is to further increase clarity in some sections (e.g. methods and discussion), and to move some pertinent information particularly on model evaluation from the SI to the manuscript. We have taken substantiative actions to make these improvements, and believe the manuscript has much improved as a result of this.

1. Abstract: It is a nice abstract, but does not provide any insights that we learn from the model application. Please add the main messages (2-3 sentences) that reflect the two main objectives of the model application:
   1. Pattern and trends
   2. Sector contributions

We agree that some text should be included in the abstract to (very broadly) summarise some key model application results. We have added the following text to the abstract:

Lines 23 - 26: *Modelled output indicates that multi-pollutant hotspots are especially prevalent across northern India and eastern China, but that surface water quality issues exist across all world regions. Trends towards water quality deterioration have been most profound in the developing world, particularly Sub-Saharan Africa and southern Asia.*

2. Methods: They are described rather concisely. Details are provided in the supplementary materials, which is nice. Nevertheless, I have four suggestions to elaborate on:
   a. Figure 1: please add the legend. The description of what colors and different arrows (dashed and solid) mean is not clear;

We have added a legend to Figure 1 (Line 115):

[Figure]

Socio-economic forcing
1. Domestic gross and net water demand (m day$^{-1}$); industry gross and net water demand (m day$^{-1}$); livestock gross and net water demand (m day$^{-1}$)
2. Population numbers; livestock numbers; urban fraction; thermo-electric power return flows (m$^3$ s$^{-1}$); excretion rates (g or cfu day$^{-1}$); mean effluent concentrations (mg l$^{-1}$ or cfu 100ml$^{-1}$)

Climate forcing
3. Air temperature (°C); precipitation (m day$^{-1}$); potential evapotranspiration (m day$^{-1}$)
4. Solar radiation (W m$^{-2}$); annual average air temperature (°C); vapour pressure (kPa); cloud cover (%); sunlight hours (hours day$^{-1}$)

Hydrological data
5. Baseflow or groundwater discharge (m day$^{-1}$); interflow (m day$^{-1}$); direct or surface runoff (m day$^{-1}$)
6. Manufacturing return flows (m$^3$ day$^{-1}$); irrigation return flows (m$^3$ day$^{-1}$); urban surface runoff (m$^3$ day$^{-1}$)
7. Water temperature (°C)

Pollutant loading data
8. Anthropogenic temperature loading (MW); TDS loading (g day$^{-1}$); BOD loading (g day$^{-1}$); FC loading (10$^6$ cfu day$^{-1}$)

b. Pollutant loadings to streams: this is well described in the supplementary information, but very concisely described in the main methods. Please add a few more sentences to tell the reader how pollutant loadings are calculated (e.g., the summary of the description from the SI). You can elaborate on the text where you mention the mass-balance approach. Here, you can briefly tell that pollutant loadings from livestock activities are simulated as a function of livestock number, excretion rates of pollutants per animal and day, and removal of pollutants during waste management practices while considering runoff from land to streams (see SI Section 1.4 for details). A similar description can be given for other sectors. And:

c. Sector contribution: you do mention sectors in the methods, but briefly. Please elaborate more on what sectors exactly include and how. For example, the irrigation sector: what does it include? Which crops? rainfed irrigation? Livestock sector: which animals? I do know that some details are in the SI. But I do feel a need to give a bit more description of the main methods of the manuscript.

We made a deliberate choice to keep the pollutant loadings section relatively short in the manuscript. This is because the pollutant routing and in-stream decay components of DynQual can also be used independent of the pollutant loadings i.e. users can define their

own pollutant loadings, based on their preferred methodology (see Section 2.1 and Figure 1). We feel this important point may be lost on the reader if there is such a strong emphasis on pollutant loading quantifications in the manuscript itself, hence our decision to largely keep this in the SI.

Nevertheless, we agree with the reviewer that some pertinent information on the estimation of pollutant loadings should indeed be included in the manuscript, and we have revised this section by more explicitly referring to the relevant sections in the SI. To address this, we have added the following text to Section 2.3:

Lines 304 – 325: *Loadings from the domestic sector are estimated by multiplying the gridded population with region-specific per capita excretion rates (SI Section 1.1, Table S1). For the manufacturing sector, a mean effluent concentration is multiplied by location-specific gridded estimates of return flows from the manufacturing sector (SI Section 1.2, Table S2). Urban surface return flows are approximated by multiplying surface runoff (simulated by PCR-GLOBWB2) with the gridded urban fraction, which are multiplied by a region-specific mean urban surface runoff effluent concentration (SI Section 1.3; Table S3). The livestock sector is sub-divided into 'intensive' and 'extensive' production systems based on livestock densities to better account for differences in the paths by which waste enters the stream network (SI Section 1.4, Table S4). Gridded livestock numbers for buffalo, chickens, cows, ducks, goats, horses, pigs and sheep are multiplied by pollutant excretion rates per livestock type and by region (SI Section 1.4, Table S5 – S7). TDS loadings from the irrigation sector are estimated by multiplying irrigation return flows simulated by PCR-GLOBWB2 with spatially-explicit mean irrigation drainage concentrations based on salinity (as indicated by electrical conductivity) over the top- and sub-soil (SI Section 1.5). Thermal effluents (heat dumps) from thermoelectric powerplants are included as point sources of advected heat by considering the temperature difference between the flows and ambient surface water temperature conditions (SI Section 1.6). Pollutant loadings from the domestic, manufacturing and intensive livestock sectors, and from urban surface runoff, are abated based on gridcell-specific wastewater practices. The proportion of pollutant loadings removed by wastewater treatment practices is estimated by multiplying the fraction of each treatment level occurring in a gridcell by the pollutant removal efficiency associated with that treatment level, as described in detail in previous work (Jones et al., 2021; Jones et al., 2022).*

As per the reviewer suggestion, we have also included a new table in the manuscript to further clarify the required input data for estimating pollutant loadings within a DynQual run (see below).

3. Downscaling: for example, some of the input data (e.g., livestock numbers) is regional, but DynQual requires the grid cell data. How did the authors go from regional to the grid cell, but also from annual to daily levels? Which model inputs require scaling (e.g., annual->daily; regional -> grid cell)? and which did not. This is not well elaborated. I suggest adding a few sentences on this in the main methods and giving more details in the SI. I suggest adding an overview table showing the list of model inputs and indicating which ones were aggregated from region to grid and from annual to daily.

We agree that an overview table that describes: 1) the list of model inputs per water use sector; 2) data source and 3) spatio-temporal resolution is a valuable addition to the manuscript, with the elaborated details per sector in the SI (sections 1.1 – 1.6).

We have added a *Table* to the manuscript (Line 361):

| Sector | Data | Source | Spatio-temporal resolution |
|---|---|---|---|
| Domestic | Population | (Lange and Geiger, 2020) | 5 arc-min; annual |
| | Excretion rates | (UNEP, 2016; Van Vliet et al., 2021) | Regional; constant |
| Manufacturing | Manufacturing return flows | PCR-GLOBWB2 (*simulated*) | 5 arc-min; daily |
| | Effluent concentrations | (UNEP, 2016; Van Vliet et al., 2021) | Global; constant |
| Urban surface runoff | Urban surface runoff | PCR-GLOBWB2 (*simulated*) | 5 arc-min; daily |
| | Effluent concentrations | (UNEP, 2016) | Regional; constant |
| Livestock | Livestock populations | (Gilbert et al., 2018) | 5 arc-min; annual |
| | Excretion rates | (Weaver et al., 2005; Wilcock, 2006; Robinson et al., 2011; Wen et al., 2017; Vigiak et al., 2019; Van Vliet et al., 2021) | Regional; constant |
| Irrigation | Irrigation return flows | PCR-GLOBWB2 (*simulated*) | 5 arc-min; daily |
| | Effluent concentrations | (Batjes, 2005) | 30 arc-min; constant |
| Power | Power return flows | (Lohrmann et al., 2019) | 5 arc-min; annual |
| | ΔT | (Van Vliet et al., 2012a) | Global; constant |

With respect to livestock numbers explicitly, these were sourced at 5 arc-min resolution directly from Gilbert et al., (2018) for a reference year of 2010. Thus, no additional downscaling was required on this data. What was required was to account for temporal changes to livestock numbers over past years, for which we had to make a coarse assumption (regional-scale percentage changes applied equally across all gridcells within that region) based on FAO data on changes to livestock numbers in the past decades (Thomson, 2003). We have added a description of this to the discussions section:

Lines 751 – 757: *A lack of temporally-explicit input data can hinder proper representation of sectors with strong intra- or inter-annual variability. For instance, notable limitations for the livestock sector are the simplified assumptions made for livestock population numbers (assumed to be constant across days of the year), changes to livestock numbers across multi-year periods (applied annually and based on regional averages) and transportation pathways to the stream network (assumed to be a function of surface runoff excluding the representation of processes that affect pollutant retention in soils).*

4. Discussion: It is very concise and to the point. Some aspects can be expanded and a few aspects can be added:
   a. Comparison with other studies: the authors do this for the pollutants that they consider. I also think that the manuscript will benefit if the authors add comparisons in terms of modeling approaches, pollution hotspots, and sector contributions (what new aspects are added in this DynQual model and what new aspects we learn from the model

application compared to other models). The authors may consider expanding the discussion (a few sentences) on comparing their pollution hotspots not only for TDS, BOD, and FC but also for other pollutants as well because pollution hotpots often match between pollutants.

We have added a short section to our discussion to compare the results from our model application (e.g. pollution hotspots) to other studies:

Lines 788 – 795: *Overall, our modelled spatial patterns in surface water quality match well with previous regional and global assessments – displaying multi-pollutant hotspots (e.g. TDS, BOD, FC) to be located across northern India and eastern China in particular (UNEP, 2016; Wen et al., 2017; Van Vliet et al., 2021). Consistent with a recent data-driven (machine learning) approach (Desbureaux et al., 2022), albeit for some different water quality constituents (e.g. total phosphorus), we find a general trend towards surface water quality improvement in development countries and deterioration in developing countries.*

b.  Implications of the limitations: any models have limitations. DynQual has as well. Examples are livestock numbers in extensive and intensive production systems that do not vary among days, excretions rates of pollutants in manure, and human waste that are constant across the days and within the regions. I understand that sources such as open defecation, and direct discharges of manure to rivers are not considered. It is fine, but this needs to be discussed. It is important to give examples of the main limitations and reflect critically on their implications on the main conclusions of the manuscript.

We have made substantial efforts to expand our section on model uncertainties to include more examples, including the example suggested by the reviewer with respect to constant livestock numbers.

Line 736 – 761: *Uncertainties in surface water quality simulations arise from a combination of uncertainties associated with quantifications of pollutant loadings (e.g. pollutant excretion, emission rates and sector-specific return flows), the quality of hydrological simulations (e.g. discharge and velocities) and the representation of in-stream processes (e.g. decay coefficients). These uncertainties are especially prevalent when modelling at large spatial extents. In-stream pollutant concentrations are sensitive to dilution capacity, thus the quality of the hydrological simulations. This issue contributes to uncertainties in simulated concentrations particularly in headwater streams. Fixed estimates of decay coefficients are assumed, which contributes to uncertainties in simulations of reactive constituents such as BOD and FC. In addition, the representation of lakes and reservoirs in DynQual is rudimentary, with total (routed) loadings instantaneously averaged over the volume of the water body assuming full mixing.*

*With respect to pollutant loading quantifications, spatial mismatches between the generation of pollutant loadings and the location of entry to the stream network (return flows) can result in the simulation of unrealistic concentrations, particularly in gridcells with very low water availability (i.e. headwater streams). This can occur where the drivers of point-source pollutant emissions (e.g. population) do not directly coincide with the location of wastewater treatment plant outlets. A lack of temporally-explicit input data can hinder proper representation of sectors with strong intra- or inter-annual variability. For instance, notable*

*limitations for the livestock sector are the simplified assumptions made for livestock population numbers (assumed to be constant across days of the year), changes to livestock numbers across multi-year periods (applied annually and based on regional averages) and transportation pathways to the stream network (assumed to be a function of surface runoff excluding the representation of processes that affect pollutant retention in soils). Locally relevant sources of pollution may also be entirely excluded, such as the lack of information on TDS emissions from mining activities and road-deicing. Similarly, pulses of pollutant loadings occurring during extreme rainfall of flood events are also overlooked, such as those associated with sewer overflows or from inundated industrial areas.*

We have also added a section to more critically evaluate the overarching purpose and applications of these type of global scale models, providing examples of the main type of research questions that DynQual can help to investigate (please see our response to Comment 3).

5. The usefulness of the model: DynQual has many useful applications (e.g., trends, patterns, future analyses, etc). The authors briefly mention this in paragraph 545. I think the authors can better emphasize how useful their model is compared to other models. For example: which scientific questions we can answer with this model that we could not answer with the previous models? The authors could add a few sentences on this in paragraph 545.

We have included some more text in the discussion to further emphasise the types of research questions DynQual can help to address, with specific reference to the results (Figures) we present in the manuscript:

Line 714 – 731: *DynQual is ambitious in its aim to model global surface water quality 1) using a consistent approach; 2) dynamically; 3) considering multiple water quality constituents; and 4) at a high spatio-temporal (i.e. 5 arc-min and daily timestep) resolution. Any model must consider the trade-offs between model complexity and availability of input datasets and data to parameterise process descriptions of the model (Weaver and Zwiers, 2000; Wen et al., 2017), and the impact of this on model scope. Being a global model, DynQual is inherently unable to accurately represent all aspects relevant to the local context. Rather, the modelling strategy is to focus on the main spatial and temporal drivers of pollution in river networks globally to facilitate first-order approximations of in-stream concentrations at high spatial (5 arc-min) and temporal (daily) resolution with global coverage. As such, DynQual allows for the investigation of research questions that only large-scale modelling efforts can address. These include, as presented in the model application section, global pollution hot- and bright- spot identification (Figures 6 - 8), the relative importance of different contributing sectors to water quality status across the globe (Figure 9) and meta-trends in surface water quality dynamics (Figures 10 – 11). The dynamic nature of DynQual can also facilitate analysis of intra- and inter- annual trends in surface water quality, and help to further enhance the understanding of the main drivers of pollution via (dynamic) sectoral attribution (Figure 12). Furthermore, this approach has particular value for simulating surface water quality in ungauged catchments, and our use of globally consistent input data facilitates meaningful comparisons across different world regions.*

6. Supplementary information: It is well written. I have three suggestions:

   a. Units: they are missed in some equations. For example, units are not included for the following variables: $R_{dom,i,n}$ and $Pop_n$ (population per km$^2$? Total population?) in equation [1], $L_{man,i,n}$ and $R_{man,i,n}$ in equation [2], $L_{urb,i,n}$ and $R_{USR,i,n}$ in equation [3]. Please also check the variables

in equations [4] and [5] and add units for every variable. This will avoid misinterpretation.

Thanks for highlighting this. We have thoroughly gone through both the manuscript and the SI to ensure that units are provided, and to improve clarity on this. We have also made some small corrections to remove inconsistencies in terms used across the two documents (e.g. in the manuscript equations we denoted pollutant using *i* but in the SI using *p* – we now consistently use *p* across both documents).

We have also added the equations for estimating removal of pollutants at wastewater treatment facilities per sector (e.g. $R_{dom,i,n}$, $R_{man,i,n}$). As these have been described extensively in previous work, we refrain from providing a full detailed description of these again here - yet we agree it is important to present the detailed equations with the terms again in the SI:

SI Lines 30 – 48: *The prevalence of wastewater collection and treatment, combined with their associated pollutant removal efficiencies, are key factors controlling subsequent delivery of pollution to surface waters (Jones et al., 2022). The fraction of pollutant loadings removed by wastewater treatment (-) are estimated for the domestic ($R_{dom,p,n}$), manufacturing ($R_{man,p,n}$) and intensive livestock ($R_{intLiv,p,n}$) sectors, and from urban surface runoff ($R_{USR,p,n}$) by multiplying the fraction of each treatment level occurring in a gridcell by the pollutant removal efficiency associated with that treatment level [1].*

$$R_{dom,p,n} = (f_{ter_n} \cdot r_{ter_p} + f_{sec_n} \cdot r_{sec_p} + f_{pri_n} \cdot r_{pri_p}) + (f_{bs_n} \cdot r_{bs_p}) + (f_{od_n} \cdot (1 - s_n))$$

$$R_{man,p,n} = f_{ter_n} \cdot r_{ter_p} + f_{sec_n} \cdot r_{sec_p} + f_{pri_n} \cdot r_{pri_p}$$

$$R_{USR,p,n} = f_{ter_n} \cdot r_{ter_p} + f_{sec_n} \cdot r_{sec_p} + f_{pri_n} \cdot r_{pri_p}$$

$$R_{intLiv,p,n} = (f_{ter_n} + f_{sec_n}) \cdot r_{sec_p}$$

[1]

*Where: f is the fraction of tertiary+ treatment ($f_{ter_n}$), secondary treatment ($f_{sec_n}$) and primary treatment ($f_{pri_n}$) within gridcell n, and r is the removal efficiency associated with tertiary ($r_{ter_{p,r}}$), secondary ($r_{sec_{p,r}}$) and primary ($r_{pri_{p,r}}$) treatment per pollutant p. $f_{bs_n}$ and , $f_{od_n}$ is the fraction of basic sanitation and open defecation, respectively, within gridcell n, $r_{bs_i}$ is the reduction in pollutant p from basic sanitation collection facilities and $s_n$ is the gridcell specific surface runoff fraction. For more detailed information about the development and implementation of gridcell specific wastewater treatment practices and their inclusion in DynQual, we refer to the previous work (Jones et al., 2021; Jones et al., 2022).*

b. Livestock activities: is the number of livestock the same per day? Is this number per km$^2$ or ha? Did you consider soil processes and associated retentions of the pollutant in soil when you calculate loadings into the streams? All this was not very clear to me. Please clarify the description of equation 4.

Gilbert et al., (2010) provide a snapshot of global livestock numbers (for the 8 livestock types considered) with a reference year of 2010. We then apply estimates of annual change (at regional spatial scale) to livestock numbers in the past and future based on statistics from the FAO to estimate livestock numbers for the other timesteps. Thus, yes, the number of livestock remains constant per day throughout the year. Soil processes and associated retentions of pollutants in soil are not explicitly accounted for, only the transport of pollution from this sector to streams via surface runoff. These are limitations which we now explicitly state in the discussion section of the manuscript (see Comment 2b & 2c), and areas for improvement in estimates of pollutant loadings from the livestock sector within DynQual.

In addition we have also added some sentences to further clarify on these points in the SI:

SI Lines 173 – 177: *Gridded livestock numbers at 5 arc-minutes are derived at the annual timescale from a global dataset for the reference year of 2010 (Gilbert et al., 2018). Thus, we do not account for intra-annual variations in livestock numbers. For the quantification of past gridded livestock numbers, a region-specific (annual) constant percentage growth change in the number of animals per livestock type is applied to all gridcells based on data from the FAO (Thomson, 2003) (Table S5).*

      c. Scaling: please elaborate on which input data required to be scaled from region to grid and from annual to daily, and how this was done.

Due to the significant overlap, we refer to our response to Comment 2a-c and Comment 3 (i.e. *Table 2*) to address this comment.

**Response to Reviewer #2**

In addition to (or as an emphasis of) the community comment by Jason Ke and the comments from Referee #1 I have the following suggestions for improving the paper.

We again thank the reviewer for their comments and the helpful suggestions. The overarching theme of these comments is to improve aspects related to: 1) the scientific basis and technical implementation of DynQual and large-scale water quality models in general; 2) improve and expand the representation of model evaluation (i.e. validation) aspects, in addition to better highlighting the limitations of the modelling approach; and 3) address the imbalance of contents in terms of model evaluation versus model demonstration in the manuscript. We have taken substantiative actions to improve upon these three points, in addition to the other issue raised by the reviewer, and believe the manuscript has vastly improved as a result of this.

1. Model description paper in GMD should focus on a detailed description of the scientific basis of a model and the technical/ numerical implementation. In the current version of the manuscript, there is a certain imbalance between the respective contents in the main paper and the supplemental material. Most of the description of model details in the supplemental material should be directly mentioned in the main paper.

We agree that more technical details should be included in the manuscript. We have categorised the changes we have made to the manuscript under three sub-headings: 1) surface water and pollutant routing; 2) pollutant loading calculations; and 3) other technical details regarding model code, data format and running time:

      a. Surface water and pollutant routing

We previously referred only to the PCR-GLOBWB2 documentation (*Sutanudjaja et al., 2018*) for these descriptions to avoid too much overlap. However, we agree that specific information regarding the routing routine should also be included here. We have included the following text to address this:

Lines 127 – 139: *The routine for surface water (and pollutant) routing follows an eight-point steepest-gradient algorithm across the terrain surface (local drainage direction) in a convergent drainage network with the lowermost cell connected to either the ocean or an endorheic basin as per PCR-GLOBWB2 (Sutanudjaja et al., 2018) and DynWat (Van Beek et al., 2012; Wanders et al., 2019). Routing within DynQual uses the kinematic wave approximation of the Saint-Venant equations with flow described by Manning's equation, solved using a time-explicit variable sub-time stepping scheme based on the minimum Courant number (Sutanudjaja et al., 2018). In the coupled configuration, surface waters are subject to water withdrawals and return flows from the domestic, industrial, livestock and irrigation sectors calculated within the water use module of PCR-GLOBWB2. A full complete model description of PCR-GLOBWB2 including detailed information on the model structure, individual modules (meteorology, land surface, groundwater, surface water routing and water use) and validation of hydrological output is available in published literature (Sutanudjaja et al., 2018).*

      b. Pollutant loadings

Please see our response to Reviewer 1 Comments 2b & 2c. The required input data for estimating pollutant emissions within a DynQual run have also been added in a Table (per the suggestion of Reviewer #1), as per our response to Reviewer 1 Comment 3.

      c. Other technical details.

We have added some more technical details to the manuscript regarding performing model runs, input and output files and estimates of run-times. More information, particularly with regards to installation and running DynQual, is available in the README.txt and model manual on our GitHub page (https://github.com/UU-Hydro/DYNQUAL):

Line 154 – 162: *As per PCR-GLOBWB2 (Sutanudjaja et al., 2018) and DynWat (Wanders et al., 2019), DynQual is written in Python 3 and is run using an initialization (.ini) file in which key aspects of the model run are defined (e.g. spatial extent, simulation period, paths to parameter and forcing files). Most input files required and all output files are in NetCDF format. Global 5 arc-min DynQual runs that are coupled with PCR-GLOBWB2 have a wall-clock time of approximately 6 hours per year when run with parallelisation, due to the requirement to use the kinematic wave routing option for higher accuracy discharge and water temperature simulations. This is approximately equivalent to the PCR-GLOBWB2 run times given by Sutanudjaja et al., (2018). DynQual runs performed in the stand-alone configuration are faster (~20%).*

2. Another large part of model description papers should be dedicated to the model verification. This aspect is quite underrepresented in the current version of the manuscript. Model verification is only presented in line 304 to 321 and one figure in the main paper. Some additional results are included in the supplemental material but without a thorough presentation, analysis and discussion of the results. These aspects need to gain much more space in the main paper as this is a central part of model description papers in GMD.

We agree that the model evaluation was under-represented in the initial submission. We have taken substantial actions, across both the manuscript and the SI, in order to address these concerns.

The actions that have been taken are summarised below. For full details, please refer to the manuscript and SI with tracked changes:

a) The model evaluation section has been made into its own sub-section "3.2 Model evaluation" and has now been separated from "3.1 Model run setup".

b) In the revised manuscript we elaborate more on the metrics we choose with respect to the aims and scope of global water quality models. Model evaluation in the manuscript (with complementary sub-sections in the SI) is now focussed on three main factors: 1) concentration classes (Figure 3); 2) KGE for water temperature; nRMSE for TDS, BOD and FC in Figure 4 (with spatial patterns in SI: Figures S3 and S4); and 3) time-series and average annual cycle plots (new Figure 5 + additional plots across different regions in SI Figures 5 - 8). These figures are also inserted into this document (see below).

Lines 391 – 403: *The overarching purpose and applications of a model, including large-scale water quality models (Beusen et al., 2015; UNEP, 2016), must be considered both for determining suitable metrics for model evaluation and for judging model performance. Given the approximations in the model, uncertainties in input data and the overall complexity in the drivers of pollutant loadings, the purpose of global water quality models is not to compute daily concentrations exactly (UNEP, 2016). The modelling strategy is thus to focus on the main spatial and temporal drivers of pollution in river networks globally to facilitate first-order approximations of in-stream concentrations. A key reason for implementing DynQual at 5 arc-minute spatial resolution is due to the marked improvement of the performance of both PCR-GLOBWB2 (e.g. discharge) (Sutanudjaja et al., 2018) and DynWat (e.g. water temperature) (Wanders et al., 2019) at finer spatial extents. These two factors have an important influence on simulated in-stream concentrations due to dilution and in-stream decay processes, respectively.*

Lines 403 – 421 : *Given these factors, combined with limitations in the observational records of surface water quality (SI Section 3.1), global water quality models have typically not been evaluated with metrics commonly used for hydrological modelling such as coefficients of determination, Nash-Sutcliffe efficiency (NSE) and Kling-Gupta efficiency (KGE) (Voß et al., 2012; Beusen et al., 2015; UNEP, 2016; Wen et al., 2017; Van Vliet et al., 2021), with the exception of water temperature simulations (Van Vliet et al., 2012b; Wanders et al., 2019). The model evaluation approach adopted for DynQual combines methods applied for the evaluation of other global water quality modelling efforts. Simulated TDS, BOD and FC concentrations are evaluated with respect to pollutant classes linked to key sectoral water quality thresholds (UNEP, 2016; Wen et al., 2017) (SI Section 3.1.2; Table S9) and statistically using normalised root mean square error (nRMSE) (Beusen et al., 2015; Van Vliet et al., 2021) (SI Section 3.2.2; SI Eq. [11]). This provides an indication of prediction errors across the different water quality constituents comparable with previous large-scale water quality assessments. Conversely, the quality of water temperature simulations is evaluated using KGE (SI Section 3.2.2; SI Eq. [10]). All four water quality constituents are also evaluated by considering long-term time-series and multi-year annual cycles at individual monitoring stations (SI Section 3.2.3), which we present for the station with the most data availability across all four constituents (see Figure 5 for a station in the Mattaponi river in the USA) and for a selection of additional monitoring stations per water quality constituent (Figures S5- S8).*

[Figure]

**Figure 3.** Differences in observed vs simulated pollutant classes for a) total dissolved solids (TDS), b) biological oxygen demand (BOD) and c) fecal coliform (FC). Pollutant classes are defined based on water use and ecological limitations, as stated by governmental and international organisations. A difference in classification level of "0" indicates the simulated pollutant class matches the observed pollutant class, while negative differences indicate that observed concentrations exceeded simulated concentrations, and vice-versa for positive differences.

[Figure]

**Figure 4.** Evaluation of model performance using the Kling-Gupta efficiency (KGE) coefficient for a) water temperature (Tw); and normalised root mean square error (nRMSE) for b) total dissolved solids (TDS), c) biological oxygen demand (BOD) and d) fecal coliform (FC) simulations. Spatial patterns in KGE for Tw (Figure S3) and nRMSE for TDS, BOD and FC (Figure S4) are displayed in SI Section 3.2.2.

[Figure]

**Figure 5.** Time-series (left) and average annual cycles (right) of observed versus simulated surface water quality as indicated by a) water temperature (Tw; oC); b) total dissolved solids (TDS; mg l-1) concentrations; c) biological oxygen demand (BOD; mg l-1) concentrations; and d) fecal coliform (FC; cfu 100ml-1) concentrations at an example water quality monitoring station. In the time-series plots, observations are indicated by blue crosses, daily simulations by grey lines and 30 day running averages by red lines. In the average annual cycles plots, blue and red lines indicated the median observed and simulated values, respectively, while the shading represents the range in values as indicated by the 10th and 90th percentiles. More examples for Tw (Figure S5), TDS (Figure S6), BOD (Figure S7) and FC (Figure S8) across different world regions are displayed in SI Section 3.2.3.

[Figure]

**KGE**

- > 0.8
- 0.4 - 0.8
- 0 - 0.4
- -0.4 - 0
- < -0.4

**Figure S3.** Spatial validation of water temperature (Tw) observations versus simulations using the Kling-Gupta efficiency (KGE) at for observation stations with > 30 observations over 1980 – 2019.

[Figure]

**Figure S4.** Spatial validation of a) total dissolved solids (TDS); b) biological oxygen demand; and c) fecal coliform (FC) observations versus simulations using the normalised root mean square error (nRMSE) at for observation stations with > 30 observations over 1980 – 2019.

[Figure]

**Figure S5.** Time-series (left) and average annual cycles (right) of observed versus simulated water temperature ($^O$C) at four selected monitoring stations. In the time-series plots, observations are indicated by blue crosses, daily simulations by grey lines and 30 day running averages by red lines. In the average annual cycles plots, blue and red lines indicated the median observed and simulated water temperature, respectively, while the shading represents the range in water temperatures as indicated by the 10th and 90th percentiles.

[Figure]

**Figure S6.** Time-series (left) and average annual cycles (right) of observed versus simulated total dissolved solids (TDS) concentrations (mg l$^{-1}$) at four selected monitoring stations. In the time-series plots, observations are indicated by blue crosses, daily simulations by grey lines and 30 day running averages by red lines. In the average annual cycles plots, blue and red lines indicated the median observed and simulated concentrations, respectively, while the shading represents the range in concentrations as indicated by the 10th and 90th percentiles.

[Figure]

**Figure S7.** Time-series (left) and average annual cycles (right) of observed versus simulated biological oxygen demand (BOD) concentrations (mg l$^{-1}$) at four selected monitoring stations. In the time-series plots, observations are indicated by blue crosses, daily simulations by grey lines and 30 day running averages by red lines. In the average annual cycles plots, blue and red lines indicated the median observed and simulated concentrations, respectively, while the shading represents the range in concentrations as indicated by the 10th and 90th percentiles.

[Figure]

**Figure S8.** Time-series (left) and average annual cycles (right) of observed versus simulated fecal coliform (FC) concentrations (cfu 100ml⁻¹) at four selected monitoring stations. In the time-series plots, observations are indicated by blue crosses, daily simulations by grey lines and 30 day running averages by red lines. In the average annual cycles plots, blue and red lines indicated the median observed and simulated concentrations, respectively, while the shading represents the range in concentrations as indicated by the 10th and 90th percentiles.

c) We have vastly elaborated our description of the model evaluation results in the manuscript, also with respect to the model aims:

[revised manuscript text omitted]

We have more critically reflected on model performance in the discussion section:

Lines 762 – 786: *Despite these uncertainties, DynQual has been demonstrated to perform with a reasonable level of performance, especially given the approximations of the model. Water temperature simulations closely match observations at daily resolution as indicated by KGE coefficients (Figure 4a), which are high across all world regions (Figure S3). Furthermore, time-series and average annual plots (Figure 5a; Figure S5) demonstrate that seasonal regimes present in observed water temperatures are well-captured by the model. Simulated TDS, BOD and FC concentrations are largely within the correct concentration classes (Figure 3) with nRMSE coefficients (Figure 4b-d) deemed reasonable considering the challenges of comparing individual (instantaneous) observed daily TDS, BOD and FC concentrations against simulated daily concentrations. Long-term time-series and average annual cycle plots for TDS (Figure 5b; Figure S6), BOD (Figure 5c; Figure S7) and FC (Figure 5d; Figure S8) show that DynQual can generally simulate in-stream concentrations within the correct range (e.g. min-max daily concentrations, 10th and 90th percentile average annual cycles), but simulations of in-stream concentrations timeseries on a daily timestep show relatively poor agreement with the observed timeseries. Observed data records also tend to display large variability in concentrations but little (systematic) seasonality, especially for BOD (Figure S7) and FC (Figure S8) concentrations. These factors have a strong influence on metrics including RMSE, but especially the other commonly-used evaluation metrics in hydrology such as the Nash-Sutcliffe efficiency (NSE) and Kling-Gupta efficiency (KGE), and hence support our decision not to evaluate model performance using these metrics. Challenges related to the observational records themselves should also be acknowledged. These can relate to, for example, artefacts in observational records (Figure S9a), issues related to instrument detection limits and/or reporting accuracies (Figure S9b) and large variability in the observation records (Figure S9c). Lastly, given the approximations of the model, the overall complexity in the drivers of pollutant loadings and input data limitations, we re-iterate that the current set-up of DynQual is not suited to simulate daily TDS, BOD and FC concentrations that correspond exactly with in-situ observational measurements.*

[Figure]

**Figure S9.** Examples of challenges associated with observation data when evaluating global surface water quality models: a) artefacts in the data; b) detection limits or reporting accuracy; c) large variability in the observed record.

d) We have moved the majority of the text on data availability to the SI (also to reduce the repetition). Furthermore, we have expanded this section with a new Figure (Figure S2) and a textual description of this figure. Table S8 has been updated in order to include more information on the length of time-s

SI Lines 302 – 303: *The number of stations with >30 and >90 measurements across the time period 1980 – 2019 and the associated number of observations are also presented (Table S8).*

**Table S8.** Number of water quality monitoring stations and measurements used for DynQual model validation.

| Water quality constituent | All stations | | Stations > 30 observations | | Stations > 90 observations | |
|---|---|---|---|---|---|---|
| | N Stations | N Observations | N Stations | N Observations | N Stations | N Observations |
| Water Temperature (Tw) | 22,990 | 841,781 | 7,312 | 729,813 | 2,194 | 474,567 |
| Total Dissolved Solids (TDS) | 31,509 | 6,809,700 | 26,615 | 6,722,775 | 10,494 | 5,921,049 |
| Biological Oxygen Demand (BOD) | 12,604 | 312,019 | 2,735 | 233,169 | 636 | 133,106 |
| Fecal Coliform (FC) | 7,917 | 246,652 | 2,263 | 213,705 | 863 | 136,961 |

SI Lines 317 – 329: *Both the number of water quality monitoring stations and the length of the observation record are highly unequally distributed across space (Figure S2). Spatial patterns are relatively consistent across all four water quality constituents. North America is by far the most data-rich world region, with 45%, 76%, 62% and 92% of all monitoring stations for Tw, TDS, BOD and FC located in this region, respectively, accounting for 39%, 32%, 50% and 83% of the total number of observations, respectively. Observations made across Western Europe account for 28%, 14% and 4% of Tw, BOD and FC of the total data availability, respectively, but just 3% of the TDS observations. Conversely, 58% of total TDS observations are from the East Asia & Pacific region, but just 14%, 7% and 3% of the Tw, BOD and FC observations, respectively. However strong spatial biases within individual regions must also be considered, particularly for TDS observations in East Asia & Pacific where >99% of these observations are from Australia. The Latin America and Caribbean region also accounts for a small but significant share of total Tw, TDS, BOD and FC observations, at 13%, 1%, 15% and 3% of total observations, respectively.*

SI Lines 331 – 337: *Data is extremely scarce across other world regions, especially when also considering the length of observation records (Figure S2). While there are some localised pockets of high data availability in different regions (e.g. TDS measurements in South Africa), publicly accessible observational data records are mostly non-existent. For example, the number of stations in Sub-Saharan Africa with >30 observations of Tw, BOD and FC is just 10, 1 and 1, respectively. When considering stations with >90 observations, these numbers drop to 6, 1 and 0. Similar patterns in data availability are observed for the Middle East and North Africa and Southern Asia regions.*

SI Lines 338 – 341: *The spatial biases in the observed data, combined with data availability issues in general (especially for BOD and FC), provide acute challenges for the evaluation of global water quality models across different world regions (Section SI 3.2).*

[Figure]

**Figure S2.** Number of surface water quality monitoring stations per world region, disaggregated by the total number of observations made at each site from 1980 - 2019. Black dots on the map display the locations of water quality monitoring stations with > 90 observations of any water quality constituent. Please note that different numbers are used on the vertical axis for bar charts displaying the number of observation stations for different world regions.

e) The need to better elucidate the scope, aims and rationale of DynQual (and by extension, other large-scale water quality models) in the manuscript is apparent to us. We have added paragraphs (in model evaluation, and in discussion) to highlight the need to consider the purpose of a particular model, and the associated model set-up, with respect to the types of research questions these set-ups can address (also see our response to Reviewer 1 Comment 5):

Lines 391 – 404: *The overarching purpose and applications of a model, including large-scale water quality models (Beusen et al., 2015; UNEP, 2016), must be considered both for determining suitable metrics for model evaluation and for judging model performance. Given the approximations in the model, uncertainties in input data and the overall complexity in the drivers of pollutant loadings, the purpose of global water quality models is not to compute daily concentrations exactly (UNEP, 2016). The modelling strategy is thus to focus on the main spatial and temporal drivers of pollution in river networks globally to facilitate first-order approximations of in-stream concentrations. A key reason for implementing DynQual at 5 arc-minute spatial resolution is due to the marked improvement of the performance of both PCR-GLOBWB2 (e.g. discharge) (Sutanudjaja et al., 2018) and DynWat (e.g. water temperature) (Wanders et al., 2019) at finer spatial extents. These two factors have an important influence on simulated in-stream concentrations due to dilution and in-stream decay processes, respectively.*

and:

Lines 714 – 735: *DynQual is ambitious in its aim to model global surface water quality 1) using a consistent approach; 2) dynamically; 3) considering multiple water quality constituents; and 4) at a high spatio-temporal (i.e. 5 arc-min and daily timestep) resolution. Any model must consider the trade-offs between model complexity and availability of input datasets and data to parameterise process descriptions of the model (Weaver and Zwiers, 2000; Wen et al., 2017), and the impact of this on model scope. Being a global model, DynQual is inherently unable to accurately represent all aspects relevant to the local context. Rather, the modelling strategy is to focus on the main spatial and temporal drivers of pollution in river networks globally to facilitate first-order approximations of in-stream concentrations at high spatial (5 arc-min) and temporal (daily) resolution with global coverage. As such, DynQual allows for the investigation of research questions that only large-scale modelling efforts can address. These include, as presented in the model application section, global pollution hot- and bright- spot identification (Figures 6 - 8), the relative importance of different contributing sectors to water quality status across the globe (Figure 9) and meta-trends in surface water quality dynamics (Figures 10 – 11). The dynamic nature of DynQual can also facilitate analysis of intra- and inter- annual trends in surface water quality, and help to further enhance the understanding of the main drivers of pollution via (dynamic) sectoral attribution (Figure 12). Furthermore, this approach has particular value for simulating surface water quality in ungauged catchments, and our use of globally consistent input data facilitates meaningful comparisons across different world regions. Given severe limitations in observational records of surface water quality, both in terms of spatial coverage and the number of observations per water quality monitoring station (SI Section 3.1), these are key strengths of DynQual. However, poor data availability is simultaneously a severe limitation for both the development of global water quality models and their evaluation.*

3. Along with a more detailed model verification, the authors also need to emphasize the discussion on possible model limitations.

We agree that a better discussion of model limitations is required in the manuscript. Please refer to our response to Reviewer 1 Comment 4b for the additional text we have added to the discussion section.

We have also taken extra steps to qualify the types of research questions that large-scale water quality models, such a DynQual, are suitable for answering (see response to Reviewer 1 Comment 5). These also tie into the overarching discussion on model limitations.

4. A much larger portion of the manuscript (lines 323-491) is dedicated to the presentation of spatial patterns and trends of (long-year global) simulation results. Section 3 is named "model demonstration" and the model evaluation part (see above) is only a small part of this section. Showing possible applications of the model is, of course, interesting but model fidelity has do be demonstrated first and in much greater detail before spatial patterns and trends can be presented. Hence, there needs to be a better balance between the "model evaluation" and the "model demonstration" part.

We agree that a model description paper should strike a better balance between "model evaluation" and "model demonstration" than was presented in the original version of the manuscript. We have taken two substantial actions to address this:

- As described in our response to Reviewer 2 Comment 2, we have now split the sub-section "Model run set-up and validation" into two sub-sections: i.e. 3.1) Model run setup; and 3.2 Model evaluation. We have expanded both of these sections, in line with our response to Reviewer 2 Comment 1 and Comment 2, respectively. Please refer to these sections for full details of the changes that have been incorporated.
- We have removed the results related to the average annual fluctuations (formally Figure 8 + associated description) from the manuscript, keeping the focus of our model demonstration on spatial patterns, sectoral attribution and trends.

Combined, we believe these actions have brought better balance to the manuscript in line with the requirements for a GMD model description paper.

5. The performance of DynQual should also be discussed in comparison to other water quality models, e.g. available catchment or regional scale models. The level of detail in terms of available input data and spatio-temporal resolution is, of course, different but the modelling community and potential users need to know what quality of output they could expect from this newly developed model compared to existing ones. Although the scope of the model might be a bit different, the interpretation of results needs to rely on the closeness to observation data (i.e. the prediction capability) and this one needs to be compared to already existing modelling approaches.

In the presented manuscript, the performance of DynQual has been compared against that of large-scale/ global water quality models that are comparable to ours – i.e. dynamic and spatially explicit (e.g. van Vliet et al., 2021; Wen et al., 2017 and UNEP 2016). Many of the other global models are lumped (e.g. country, or basin) and static (e.g. annual, decadal), also designed to simulate pollutant loads and not in-stream concentrations. This also

complicates meaningful model comparison. We have added text to section "4 Discussion, conclusions and future work" to elaborate on this topic:

Lines 815 – 818: *Meaningful comparisons to other surface water quality models are challenging due to the high diversity in terms of: 1) spatial extent (e.g. lumped vs. distributed); 2) temporal resolution (e.g. daily vs. monthly vs. annual vs. decadal); and 3) water quality constituent and reporting form (e.g. loads vs. concentrations).*

We also find difficulties into statistically comparing the performance of an uncalibrated global water quality model (with global parameterisation and input data sets) to watershed-specific water quality models. These two types of models have fundamentally different purposes. Watershed-scale models can incorporate locally relevant input data and processes which are impossible to meaningfully represent in global approaches given data limitations. Watershed models are parameterized for specific local conditions and typically are calibrated based upon observation data of good quality and record. We have included additional text to specifically allude to the key differences between watershed vs. global surface water quality models in the manuscript:

Lines 818 – 824: *Similarly, watershed-scale surface water quality models are constructed for different purposes than large-scale (continental to global) surface water quality models. These watershed models can better incorporate locally relevant input data and processes, are parameterized for local conditions and typically have data of good quality and record length for calibration and validation – which facilitates higher precision and accuracy in both hydrological and water quality simulations. However, these models are reliant upon detailed local knowledge which is severely lacking for many (particularly ungauged) catchments worldwide (e.g. large parts of Africa).*

As detailed in our response to Reviewer 1 Comment 5, we find it important to specifically detail the usefulness of DynQual with respect to the types of scientific questions we can use it to address. We also have added text to the final paragraph of the discussion to reflect on the key aim of global water quality models:

Lines 825 – 827: *Despite their limitations, process-based large-scale water quality models can facilitate first-order assessments of global water quality dynamics that are consistent across both space and time, such as those demonstrated in the model application section of this study.*

6. It is unclear, why the normalized RMSE was chosen as the sole performance measure for model evaluation. Other performance measures such as the Nash-Sutcliffe efficiency (NSE) or the Kling-Gupta efficiency (KGE) are much more common in model benchmarking and allow a clearer interpretation of the model performance, e.g. NSE lower or greater than zero. In addition, KGE even combines different aspects of model performance.

Please refer to our response to Reviewer 2 Comment 2 for the detailed summary of the model evaluation metrics we now use (and include in the main manuscript). With respect to not include performance metrics such as NSE and KGE, please especially see our critical reflection on model performance that we have added to the discussion section (Reviewer 2 Comment 2d).

We also note that no other global water quality models have used these metrics for evaluating model performance, with the exception of water temperature models (for which we now also use KGE).

> 7. It is mentioned that DynQual can be run in two modes: (1) coupled to PCR-GLOBWB2 and (2) in offline-mode with any other hydrological model. Please describe in more detail the technical aspects of the coupling between DynQual and PCR-GLOBWB2 as well as the (technical) requirements for using DynQual with other hydrological models, e.g. what are the required input data from the hydrological model and in which form they need to be provided to DynQual (e.g. netCDF files and their format).

We have added clarification on the specific input data required to run DynQual in the stand-alone configuration (also note this information is contained in Figure 1):

Lines 124 – 126: *"…either: 1) in a stand-alone configuration with specific discharge (i.e. baseflow, interflow and direct runoff in m day$^{-1}$) fed from any land surface or hydrological model."*

We have also clarified on input and output data formats:

Lines 156 – 157: *Most input files required and all output files are in NetCDF format.*

We have also added text to elaborate on the coupling with PCR-GLOBWB2:

Lines 134 – 136: *In the coupled configuration, surface waters are subject to water withdrawals and return flows from the domestic, industrial, livestock and irrigation sectors calculated within the water use module of PCR-GLOBWB2.*

> 8. Time-stepping: Some lower bounds for the sub-daily time steps are mentioned (line 184ff). Does the model ensure that numerical stability criteria (Courant number, Peclet number) are met for the reactive transport equations?

We have added additional information to the manuscript regarding the time-steps (and, more generally, the routine for surface water and pollutant routing):

Line 127 – 134: *The routine for surface water (and pollutant) routing follows an eight-point steepest-gradient algorithm across the terrain surface (local drainage direction) in a convergent drainage network with the lowermost cell connected to either the ocean or an endorheic basin as per PCR-GLOBWB2 (Sutanudjaja et al., 2018) and DynWat (Van Beek et al., 2012; Wanders et al., 2019). Routing within DynQual uses the kinematic wave approximation of the Saint-Venant equations with flow described by Manning's equation, solved using a time-explicit variable sub-time stepping scheme based on the minimum Courant number (Sutanudjaja et al., 2018).*

Please also refer to our response to Reviewer 2 Comment 1.

9. Equation 9 for nRMSE: Please double-check this equation, the square is missing

Thank you for catching this issue. This has been corrected (SI Line 400).

**Response to Community Comment #1:**

This manuscript developed a global water quality model DynQual V1.0 and interpreted its results for TDS, BOD, and FC. Overall this manuscript is well-written with good-quality figures. Model results regarding the spatial patterns of concentration and temporal trends by region and economic development are interesting. However, there are some concerns about the model evaluation.

We again thank Jason Ke for their comments and the helpful suggestions.

1. There seems no description of model calibration. How was the calibration done for the global water quality model? Is it a simultaneous calibration for both hydrology (discharge) and water quality (Tw, TDS, BOD, FC), or a two-step calibration strategy with discharge calibrated first followed by water quality calibration? Since the author mentioned that discharge was very important for model results (Supplement, Line 295), I would assume the discharge has to be well-calibrated before modeling water quality.

*DynQual*, in addition to the underlying model *PCR-GLOBWB2* (Sutanudjaja et al., 2018) and the original water temperature model *DynWat* (Wanders et al., 2019), are uncalibrated. We have now clarified this in the manuscript:

Lines 365 – 371: *As per PCR-GLOBWB2 (Sutanudjaja et al., 2018), in addition to the original water temperature model DynWat (Wanders et al., 2019), no calibration was performed. The process-based nature and global scale of DynQual, combined with strong spatial biases in observations (Figure S2) and the large number of parameters that need to be estimated, complicate meaningful calibration. In addition, uncalibrated physical models can theoretically be applied in ungauged basins without loss of performance and are more preferable for global change assessments with different climatic and socio-economic scenarios (Hrachowitz et al., 2013; Wanders et al., 2019).*

2. The model evaluation that is very important to the model development paper seems underdeveloped. It is essential to evaluate the model performance before the model result interpretation. For example, it is ideal to evaluate model performance whenever data are available. For example, there are 27,238 stations with TDS data in the Supplement. Perhaps the author could do the following evaluation regarding 1) spatial pattern of mean concentration (e.g., model mean vs. data mean from the station with high data availability); 2) temporal dynamics regarding seasonal fluctuations and long-term trends (e.g., Fig 11, add data points to the temporal trend plots to evaluate if the model could reproduce the long-term trends)

Please refer to our detailed response to Reviewer 2 Comment 2 for the substantial changes that have been made to the model evaluation section of our submission.

3. what is a good nRMSE value? It would be beneficiary to add the Nash–Sutcliffe model efficiency coefficient (NSE) which is a widely used dimensionless metric in hydrology and water quality literature.

Please refer to our response to Reviewer 2 Comment 6 (and, by extension, Reviewer 2 Comment 2) for our discussion on model evaluation metrics, including our choice on the use of different evaluation metrics.

4. this manuscript in general lack literature discussion or comparison in terms of model performance (e.g., Figure 3), for example, what is other water quality model performance in terms of nRMSE? There might be few global scale water quality models. But I guess it could be useful to add a few comparisons with other watershed-scale water quality models.

Please refer to our detailed response to Reviewer 2 Comment 5 for our discussion on model comparisons.

5. Line 200, can the decay coefficient be specified by the user?

Yes, decay coefficients can be specific by the user directly in the source code. In the manuscript, we present the way this has currently been implemented within DynQual (based off existing global water quality modelling work). These are displayed in Equations 4 (BOD) and 5 (FC), with parameter values for FC in Table 1. We have clarified this in the manuscript:

Lines 283 – 284: *Parameter values, including decay coefficients, can alternatively be defined by the user directly in the source code.*

6. Line 220, is it a constant background concentration or a time-varying background concentration through each timestep?

We use a constant background concentration for TDS. This has been clarified in the manuscript:

Line 251: *"…and are applied as a constant background concentration."*

While time-varying background concentrations could technically be implemented into DynQual easily, the availability of data (especially at the global scale) to determine time-varying background concentrations is a limitation. Other sources of pollutant loadings to the surface water network are time-varying, for example those related to highly seasonal irrigation regimes.

We also specifically now specifically mention the use of constant background concentrations based on minimum observations as a limitation (and as a potentially reason why TDS concentrations in coastal zones are not well simulated by DynQual):

Lines 462 – 465: *Simulated TDS concentrations are typically lower than observations in many locations that are proximate to the coastline, presumably due to a combination of background TDS concentrations based upon minimum observations (and applied constantly) and DynQual not accounting for the influence of saltwater intrusion.*

7. what was the computational time to run for 1-year simulation?

This largely depends on: 1) model configuration (Figure 1) and 2) the target spatial extent.

Global 5 arc-min DynQual runs that are coupled with PCR-GLOBWB2, as done in this work, have a wall-clock time of approximately 6 hours per year, when run with parallelization. This is due to the requirement to run with the kinematic wave routing option (see our response to Reviewer 2 Comment 1). This is a more computational demand routing equation than e.g. travel-travel time characteristic routing option (Sutanudjaja et al., 2018), but provides greater realism which is needed for higher accuracy discharge and water temperature simulations. Our calculation times are more or less equivalent to the PCR-GLOBWB2 run times given in Sutanudjaja et al., (2018) using kinematic wave routing. So called "offline" global DynQual runs, which use hydrological input (i.e. baseflow, interflow and direct runoff) as an external forcing, are somewhat (~20%) quicker.

The parallelization strategy for global PCR-GLOBWB2 - DynQual involves dividing the model domain into 53 groups of river basins that run independently of each other as 53 separate processes (see Sutanudjaja et al., 2018 for more details). The length of time it takes for each individual process to run is dependent on the size of the land mask (i.e. the number of pixels). Thus, for global parallelization runs, the computational time to run for a 1-year simulation is equal to that of the largest land mask. Yet, PCR-GLOBWB2 - DynQual does not necessarily need to be run at the global extent – users could alternatively select a particular river basin(s) or define their own land mask. The self-contained example for the Rhine basin that we provide (https://zenodo.org/record/7027242#.Y8fHOhfMJPY) (this link was also included in the manuscript) takes approximately 45 minutes to run for 1 year.

This is now clarified in the text:

Lines 158 – 162: *Global 5 arc-min DynQual runs that are coupled with PCR-GLOBWB2 have a wall-clock time of approximately 6 hours per year when run with parallelisation, due to the requirement to use the kinematic wave routing option for higher accuracy discharge and water temperature simulations. This is approximately equivalent to the PCR-GLOBWB2 run times given by Sutanudjaja et al., (2018). DynQual runs performed in the stand-alone configuration are faster (~20%).*

8.  Supplement Line 295, does it mean reaction is underestimated compared to discharge (dilution)?

The current model set-up for DynQual emissions module is to quantify loadings at the individual gridcells where the pollution activity is located (e.g. populations), with loadings entering the surface water network at the same location (which are subsequently routed through the surface water stream network based on the flow direction (routing) map; Figure 2). This approach is advantageous in that we can use globally available datasets for estimating pollutant emissions (e.g. populations). The ability to use globally consistent input data is important for DynQual runs made at the global scale (in line with the focal point of this paper), whereby we want to facilitate meaningful comparisons across different world regions in line with our key research questions (see our response to Reviewer 1 Comment 5).

This approach has the disadvantage of spatial mismatches between the generation of pollutant loadings and the actual locations where loadings enter the stream network occur. Simulated concentrations in gridcells with low water availability – i.e. headwater streams - are particularly sensitive to this, and concentrations in these gridcells are typically masked in global water quality studies. Similarly, point sources of pollution (e.g. wastewater treatment plants) typically discharge into higher-order streams, the location of which might not exactly

coincide with population density. We have now explicitly acknowledged these limitations in the revised text of the manuscript (see our response to Reviewer 1 Comment 4b & Reviewer 2 Comment 3).

DynQual tracks the mass of pollutants (not concentrations) in discrete gridcells per timestep (Figure 1). Thus, these masses are transported downstream to higher-order streams whereby the pollutant mass will be more feasible with respect to the dilution capacity – hence the better model performance in gridcells with greater water availability.

9. Supplement Line 300, what is high data availability, and how many data points during 1980-2019?

"High data availability" is relative and this is especially true when it comes to water quality observations, which are highly limited in geographical space and fragmented across time (see e.g. Figure S2). We consider >30 measurements between 1980 – 2019 for plotting spatial patterns in KGE for water temperature (Figure S3) and nRMSE for TDS, BOD and FC (Figure S4). For the time-series plots (Figure 5, Figure S5 – S9) the stations we present with very high data availability in the context of the observed data record (~99$^{th}$ percentile of the available monitoring stations). Conversely, the evaluation of concentration classes (Figure 3) uses data from all available monitoring sites. We have clarified these points in the relevant places in the manuscript.

10. Supplement Line 305, Figure S3 (b, c) what are the nRMSE and NSE values for these two rivers? It seems that the model overestimated a lot for peaks.

For Figure S3b, the nRMSE is 0.15, with a mean observed concentration of TDS 399 mg l$^{-1}$ (349 observations) and a mean simulated concentration of TDS 388 mg l$^{-1}$. For Figure S3c, the nRMSE is 0.24, with a mean observed concentration of TDS 25 mg l$^{-1}$ (468 observations) and a mean modelled concentration of TDS 24 mg l$^{-1}$. Please note that, as part of the efforts to improve the model evaluation, use of these specific stations has changed.

We want to explicitly include some time-series plots in our evaluation to demonstrate the dynamic nature of the model in providing transient simulations. Stations for which results are shown are based on the high data availability and the long-time series of observations. As detailed in our response regarding model evaluation in Reviewer 2 Comment 2, comparing individual (instantaneous) observed concentrations vs. simulated concentrations, however, comes with challenges at large scales – particularly when it comes to minimum and maximum concentrations. Exactly replicating in-stream concentrations is also outside the scope of global water quality models:

Lines 394 – 404: *Given the approximations in the model, uncertainties in input data and the overall complexity in the drivers of pollutant loadings, the purpose of global water quality models is not to compute daily concentrations exactly (UNEP, 2016). The modelling strategy is thus to focus on the main spatial and temporal drivers of pollution in river networks globally to facilitate first-order approximations of in-stream concentrations. A key reason for implementing DynQual at 5 arc-minute spatial resolution is due to the marked improvement of the performance of both PCR-GLOBWB2 (e.g. discharge) (Sutanudjaja et al., 2018) and DynWat (e.g. water temperature) (Wanders et al., 2019) at finer spatial extents. These two*

*factors have an important influence on simulated in-stream concentrations due to dilution and in-stream decay processes, respectively.*

We refrain from calculating NSE values, as per our response to Reviewer 2 Comment 6 (and, by extension, Reviewer 2 Comment 2), and note that NSE has not been used to evaluate the performance of any global water quality model to date.